# Classic but Everlasting: Traditional Gradient-Based Algorithms Converge Fast Even in Time-Varying Multi-Player Games

**Yanzheng Chen & Jun Yu** [*]
Department of Automation
University of Science and Technology of China
Hefei 230026, China
`snbcyz@mail.ustc.edu.cn, harryjun@ustc.edu.cn`

## Abstract

Last-iterate convergence behaviours of well-known algorithms are intensively investigated in various games, such as two-player bilinear zero-sum games. However, most known last-iterate convergence properties rely on strict settings where the underlying games must have time-invariant payoffs. Besides, the limited known attempts on the games with time-varying payoffs are in two-player bilinear time-varying zero-sum games and strictly monotone games. By contrast, in other time-varying games, the last-iterate behaviours of two classic algorithms, i.e., extra gradient (EG) and optimistic gradient (OG) algorithms, still lack research, especially the convergence rates in multi-player games. In this paper, we investigate the last-iterate behaviours of EG and OG algorithms for convergent perturbed games, which extend upon the usual model of time-invariant games and incorporate external factors, such as vanishing noises. Using the recently proposed notion of the tangent residual (or its modifications) as the potential function of games and the measure of proximity to the Nash equilibrium, we prove that the last-iterate convergence rates of EG and OG algorithms for perturbed games on bounded convex closed sets are $O(1/\sqrt{T})$ if such games converge to monotone games at rates fast enough and that such a result holds true for certain unconstrained perturbed games. With this result, we address an open question asking for the last-iterate convergence rate of EG and OG algorithms in constrained and time-varying settings. The above convergence rates are similar to known tight results on corresponding time-invariant games.

## 1 Introduction

This paper discusses learning in time-varying multi-player games converging to monotone games. Monotone games are a class of multi-player games (Rosen (1965)) including a wide range of important games, including two-player zero-sum games, convex-concave games, $\lambda$-cocoercive games (Lin et al. (2020)), zero-sum polymatrix games (Anagnostides et al. (2023); Cai & Daskalakis (2011); Cai et al. (2016); Daskalakis & Papadimitriou (2009)), and zero-sum socially-concave games (Even-dar et al. (2009)). Due to their wide applications, a vast literature on finding methods to approximate their Nash equilibrium actions has been produced recently.

An important part of this literature is on last-iterate behaviours of well-known algorithms, and this topic has gained much research interest in recent literature. These algorithms include extra-gradient (EG) methods (Cai et al. (2022); Feng et al. (2023); Monteiro & Svaiter (2010)), optimistic gradient (OG) methods (Cai et al. (2022); Feng et al. (2023)), negative momentum methods (Feng et al. (2023)) and weights update methods (Lin et al. (2020), proposed in Arora et al. (2012)). Although last-iterate convergence performance is a challenging topic, yet in the case of monotone games, last-iterate convergence has been discussed in many papers. Last-iterate convergence of the EG algorithm in general monotone time-invariant games has been proven in Hsieh et al. (2019) and Popov

---

[*]Corresponding Author.

(1980), and its convergence rate has proved to be $O(1/\sqrt{T})$ in unconstrained settings (Golowich et al. (2020a)) and general settings (Cai et al. (2022)). A recent result Wei et al. (2021) shows that there even always exists a linear convergence rate whose value depends on the problem.

Despite the known considerable works, most recent literature on multi-player games is based on the assumption that the underlying repeated games are time-invariant. Nevertheless, time-invariant games are unrealistic in many real-life applications (Cardoso et al. (2019); Duvocelle et al. (2023); Mai et al. (2018)), and more realistic learning settings should allow the underlying cost functions of games to change with time. Games with such settings are called "time-varying games". Topics on time-varying games have gained popularity since several years ago. Duvocelle et al. (2023), Feng et al. (2023) and Zhang et al. (2022) are the first known successful attempts to solve problems about last-iterate convergence in time-varying games. However, Duvocelle et al. (2023) requires decreasing step size and does not provide a convergence rate but only provides that the probability of convergence is 1. By contrast, decreasing step size is unnatural since such a requirement considers new information to be decreasingly important instead of equally or increasingly as normally expected Lin et al. (2020). Besides, that result only illustrates last-iterate convergence in two-player zero-sum bilinear games in the unconstrained case, which is too special compared with general multi-player games and constrained games. Feng et al. (2023) does not require decreasing step sizes, but an unconstrained set of actions is still necessary.

Considering the limits of known research mentioned above, we conclude that the last-iterate convergence behaviours of time-varying games are still far from fully understood and none of the existing results on time-varying games provide satisfactory answers on whether and how fast an algorithm converges except in unconstrained two-player bilinear games. Hence, an open question arises naturally:

> *Will learning algorithms such as the extra gradient or optimistic gradient algorithms exhibit a last-iterate convergence rate in time-varying games with a constant step size?*

**Our contribution.** Motivated by known results, we first prove that there exist convergence rates on certain bounded and unconstrained multi-player games with a time-varying cost function for each player where the vector of cost functions $f_t$ varies with time in the following way (called convergent perturbed game): $f_t = f_\infty + g_t, \lim_{t \to \infty} g_t = 0$. In this paper, we show the following results:

Assuming that $\sum_{t=0}^{\infty} \max \|G_t\| \leq \infty$ where $G_t = \nabla g_t$, with $z^*$ defined as the Nash equilibrium of $\mathcal{G}$, we prove that the last-iterate convergence rate is $\max\{O(1/\sqrt{T}), O(\sqrt{\sum_{k=T/2}^{\infty} \max \|G_k\|}), O(\max \|G_T\|)\}$ for general convex bounded cases for the EG algorithm, $\max\{O(1/\sqrt{T}), O(\max_{k \geq T/2} L_{G_k}), O(\sqrt{\sum_{k=T/2}^{\infty} \max \|G_k\|})\}$ with $L_G$-smooth perturbing and $L$-smooth limits when $G_k(z^*) = 0$ for the EG algorithm, $\max\{O(1/\sqrt{T}), O(\sqrt{\sum_{k=T/2}^{\infty} \max \|G_k\|}), O(\max \|G_T\|)\}$ for general convex bounded cases for the OG algorithm and $\max\{O(1/\sqrt{T}), O(\sqrt{\sum_{k=T/2}^{\infty} \max \|G_k\|}), O(\max \|G_T\|), O(L_{G_{T-1}})\}$ with $L_G$-smooth perturbing and $L$-smooth limits when $G_k(z^*) = 0$ for the OG algorithm, where $T$ refers to the time. For the extra gradient and optimistic gradient algorithms, that convergence rate is approximately equal to the known tight bound for time-invariant games. Those results mean that both algorithms are robust to disturbance.

Our last-iterate results of EG and OG generalize that of prior work Cai et al. (2022) and Feng et al. (2023) significantly, where convergence rates of EG and OG have only been proved in two-player bilinear time-varying games and monotone time-invariant games, respectively. Besides, our method provides a partial answer to the open question about constraint games (Feng et al. (2023)) from a different view.

**Organization.** Section 2 defines the game problems in this paper and describes the main preliminaries applied to solve these problems. Sections 3 and 4 introduce the theoretical results on the last-iterate convergence of errors defined as tangent residuals in the EG and OG algorithms and provide their proof sketches, respectively. Section 5 illustrates the convergence performances of the EG and OG algorithms with numerical experiments. Section 6 concludes the paper with additional discussions and proposals on directions for further investigations.

## 1.1 RELATED WORK

**Related works on time-varying games.** In recent years, time-varying games have become the research interest of a group of researchers. Most of the papers have focused only on correlated equilibrium or time-average convergence. The work most closely related to ours is Duvocelle et al. (2023), investigating strictly monotone games. However, even two-player zero-sum bilinear games are not strictly monotone ($\langle F(z), z - z' \rangle = 0$). Feng et al. (2023) discussed two-player zero-sum bilinear games in unconstrained settings and obtained the first known result on the last-iterate convergence in time-varying games. Another related paper, Anagnostides et al. (2023), focused on correlated equilibria for the multi-player time-varying case. Other known results, such as Zhang et al. (2022), are focused on regret bounds (two-player) in time-varying bilinear saddle-point problems parameterized by the similarity of the payoff matrices and the equilibria of these games. Another result related to time-varying games, Harris et al. (2023), is about meta-learning in games, where each game can be repeated for multiple iterations consisting of settings in which many similar games need to be solved together. Cardoso et al. (2019) provides an optimal solution based on Nash equilibrium regret. In this paper, we provide a different perspective on time-varying games.

**Previous investigation on convergence rates of EG, OG and other related algorithms.** EG and OG algorithms have a long history (Korpelevich (1976); Popov (1980)) and the convergence rates of EG and OG in time-invariant games have been thoroughly investigated. In recent years, unconstrained strongly monotone games and unconstrained bilinear games have been shown to have linear convergence rates for EG, OG, and other variants in Daskalakis et al. (2018). Later papers have proved their asymptotic convergence (Daskalakis & Panageas (2019)). The convergence property of EG on concave games has been investigated in Monteiro & Svaiter (2010) (last-iterate) and Nemirovski (2004), and the convergence property in special non-concave games has been proved in Mertikopoulos et al. (2019).

## 2 NOTATIONS AND PRELIMINARIES

In this paper, we focus on time-varying games that converge to smooth monotone games in the following form: $\mathcal{G} = \{[[N]], \{\mathcal{Z}^{(i)}\}_{i \in [[N]]}, \{f^{(i)}\}_{i \in [[N]]}\} =: \{\mathcal{N}, \mathcal{Z}, f\}$ where $[[N]] := \{1, 2, \cdots, N\}$ is the set of players, $\mathcal{Z} \subseteq \mathbb{R}^n$ is a closed convex set and the action set of players, $\{f^{(i)}\}_{i \in [[N]]}$ are the cost functions of corresponding players and $D_{\mathcal{Z}} = \max_{x_1, x_2 \in \mathcal{Z}} \|x_1 - x_2\|$. In addition, $z$ consists of actions of players, and throughout the paper, with $\nabla$ defined as the symbol of the gradient feedback vector of the vector function, i.e., $\nabla f := (\nabla_{z^{(1)}} f^{(1)}, \cdots, \nabla_{z^{(N)}} f^{(N)})$, for simplicity, $\nabla f$ is denoted as $F$ and $\nabla g$ is denoted as $G$; $f_k$, $g_k$, $F_k$ and $G_k$ refer to $f$, $g$, $F$ and $G$ at time $k$; and subscript $k_z$ refers to the time $k$ when $z$ is applied. $\max \|G_k\|$, $\max \|F_\infty\|$, $\max \|G_{t^*}\|$ are maximums defined on $\mathcal{Z}$ if $\mathcal{Z}$ is bounded, on $\{z_i | i \in \mathbb{N}$ or $i - \frac{1}{2} \in \mathbb{N}\}$ if $\mathcal{Z}$ is unbounded with the EG algorithm, and on $\{z_i | i \in \mathbb{N}\} \cup \{w_i | i \in \mathbb{N}\}$ if $\mathcal{Z}$ is unbounded with the OG algorithm. We define the Nash equilibrium of these time-varying games as the Nash equilibrium of the limits of these time-varying games.

**Time-invariant monotone games, time-varying games and their Nash equilibria.** In this part, we introduce our model of time-varying games and the definition of Nash equilibria on them. To introduce them, we define time-invariant monotone games in Definition 1 and the sufficient and necessary condition of Nash equilibria in Lemma 1 at first. Lemma 1 shows that a Nash equilibrium of the game $\mathcal{G}$ is equivalent to a solution of the variational inequality of monotone operator $F$.

**Definition 1** *(Rosen (1965)) A game $\mathcal{G}$ is monotone if $\forall x_1, x_2 \in \mathcal{Z}$, $\langle F(x_1) - F(x_2), x_1 - x_2 \rangle \geq 0$.*

**Lemma 1** *(Facchinei & Pang (2007)) For a monotone game $\mathcal{G}$, an action $z^*$ is a Nash equilibrium of $\mathcal{G}$ if and only if $\forall z \in \mathcal{Z}$, $\langle F(z^*), z^* - z \rangle \leq 0$.*

Based on the preliminaries in time-invariant games above, we define the time-varying perturbed games investigated in our paper.

**Definition 2** (Convergent perturbed games) *A convergent perturbed game consists of an infinite sequence of games with cost functions satisfying $\{f_t(z)\}_{t=0}^{\infty} \subset \mathbb{R}^n$ where $\lim_{t \to \infty} f_t(z) = f(z)$ for*

*a certain $f(z)$. An equivalent definition is that there exists a sequence of cost functions $\{g_t(z)\}_{t=0}^{\infty} \subset \mathbb{R}^n$ and a function $f(z) \in \mathbb{R}^n$ such that $f_t(z) = f(z) + g_t(z)$ in the same infinite sequence of games. For simplicity, $f(z)$ above is denoted as $f_\infty(z)$.*

Specifically, we focus on time-varying games converging to smooth monotone games, i.e., $F_\infty(z)$ of the games are monotone and $L$-Lipschitz. The definition of smooth games is the following.

**Definition 3** *(Rosen (1965)) $\mathcal{G}$ is $L$-smooth if $F$ is $L$-Lipschitz, i.e., $\forall x_1, x_2 \in \mathcal{Z}$, $\|F(x_1) - F(x_2)\| \leq L\|x_1 - x_2\|$.*

We define the Nash equilibrium of these time-varying games as the Nash equilibrium of the limits of these time-varying games.

**Definition 4** *$z^*$ is a Nash equilibrium of a convergent time-varying game $\mathcal{G}$ if $z^*$ is a Nash equilibrium of $\mathcal{G}_\infty := \{\mathcal{N}, \mathcal{Z}, f_\infty\}$.*

In convergent time-varying games, we modify Lemma 1 as the following lemma.

**Lemma 2** *For a game $\mathcal{G}$ converging to a monotone game, an action $z^*$ is a Nash equilibrium of $\mathcal{G}$ if and only if $\forall z \in \mathcal{Z}$, $\langle F_\infty(z^*), z^* - z \rangle \leq 0$.*

Note that all games converging to monotone games have at least one Nash equilibrium if $\mathcal{Z}$ is bounded, while these games with unbounded $\mathcal{Z}$ may also have a Nash equilibrium under certain circumstances (Facchinei & Pang (2007)). Throughout the paper, we apply the following assumption on the existence of the Nash equilibrium.

**Assumption 1** (Existence of the Nash equilibrium) Any time-varying game $\mathcal{G}$ involved has at least one Nash equilibrium.

**Learning algorithm in games.** In this part, we introduce two kinds of learning algorithms investigated here: extra gradient and optimistic gradient algorithms. Both algorithms are proved to be last-iterate convergent in time-invariant games (Cai et al. (2022)). This inspires the time-varying variant of the two algorithms in general time-varying multi-player games defined in Eq. (1) and Eq. (2) and illustrated in Figure 1, where the projection operator $\Pi$ is defined as $\Pi_{\mathcal{Z}}(z) = \operatorname{argmin}_{z' \in \mathcal{Z}} \|z - z'\|$.

The extra gradient algorithm is defined as follows:

$$z_{k+\frac{1}{2}} = \Pi_{\mathcal{Z}}\left[z_k - \eta F_k(z_k)\right], z_{k+1} = \Pi_{\mathcal{Z}}\left[z_k - \eta F_k\left(z_{k+\frac{1}{2}}\right)\right] \tag{1}$$

where the step size $\eta > 0$, $z_0$ is an arbitrary point in $\mathcal{Z}$, $z_k$ is the vector consisting of actions of all players at the time $k$, and $z_{k+\frac{1}{2}}$ is a vector used to calculate the actions of players at the time $k+1$.

The optimistic gradient algorithm is defined as follows:

$$w_{k+1} = \Pi_{\mathcal{Z}}\left[z_k - \eta F_k(w_k)\right], z_{k+1} = \Pi_{\mathcal{Z}}\left[z_k - \eta F_k(w_{k+1})\right] \tag{2}$$

where the step size $\eta > 0$, $z_0$ and $w_0$ are arbitrary points in $\mathcal{Z}$, $w_k$ is the vector consisting of actions of all players at the time $k$, and $z_k$ is a vector used to calculate the actions of players at the time $k+1$.

Both algorithms have a long history. The extra gradient algorithm was proposed in Korpelevich (1976). The form of time dependence in our paper originates from Feng et al. (2023). The optimistic gradient algorithm was proposed in Popov (1980). Several versions of the OG algorithm exist in the known literature (Cai et al. (2022); Hsieh et al. (2019)). The version above is applied for convenience of analysis on time-varying $F$.

**Tangent residual as the measure of the proximity to Nash equilibria.** In former results, the common measures of the proximity of actions to Nash equilibria of games are the gap functions (Nemirovski (2004); Hsieh et al. (2019)) defined as follows.

**Definition 5** (Gap and total gap functions) *For time-varying games converging to monotone games, measures of the proximity of an action profile $z \in \mathcal{Z}$ to Nash equilibrium include their gap functions and total gap functions. For a fixed parameter $D$, the gap function is defined as $P_{\mathcal{G},D}(z) =$*

$\max_{z' \in \mathcal{Z} \cap B(z,D)} \langle F_{\mathcal{G}}(z), z - z' \rangle$, where $B(z, D)$ is a ball with radius $D$ centered at $z$. The total gap function is defined as $T_{\mathcal{G},D}(z) = \sum_{i \in [[N]]} \left( f^{(i)}(z) - \min_{z'^{(i)} \in \mathcal{Z}^{(i)} \cap B(z^{(i)}, D)} f^{(i)}(z'^{(i)}, z^{(-i)}) \right)$, where $z^{(-i)}$ consists of actions of all players except Player $i$.

However, gap functions are far from monotone. In fact, they are not monotone even in time-invariant monotone games (Cai et al. (2022)). This causes inconvenience in our analysis. To solve that problem, we adopt the tangent residual from Cai et al. (2022) as the measure of the proximity to the Nash equilibrium, since it is, or is related to, monotone non-increasing functions of time in time-invariant monotone games. The tangent residual in our settings is defined as follows. Though, due to the cost function being time-varying in our settings, the tangent residual and its relative function are only not increasing much with time instead of monotone non-increasing functions of time. The details of those functions are discussed in Sections 3 and 4.

**Definition 6** (Tangent residual in convergent games) *For any closed convex set $\mathcal{Z}$ and operator $F : \mathcal{Z} \to \mathbb{R}^n$, define $N_{\mathcal{Z}}(z)$ as the normal cone of $z$, $\hat{N}_{\mathcal{Z}}(z) = \{z | z \in N_{\mathcal{Z}}(z), \|z\| \le 1\}$ and $J_{\mathcal{Z}}(z) := \{z\} + T_{\mathcal{Z}}(z)$, where $T_{\mathcal{Z}}(z) = \{z' \in \mathbb{R}^n : \langle z', a \rangle \le 0, \forall a \in N_{\mathcal{Z}}(z)\}$ is the tangent cone of $z$. The tangent residual of $\mathcal{G}$ is*

$$r_{F,\mathcal{Z}}^{tan}(z) = \left\| \Pi_{J_{\mathcal{Z}}(z)} [z - F(z)] - z \right\|$$

To show the reasonability of using the tangent residual in our settings, we also show the relationship between the tangent residual and the gap and total gap functions in the following lemma. Specifically, it shows that gap functions are not much larger than tangent residuals, which means that tangent residuals converge no more slowly than gap functions. As a result, the results shown in this paper are applicable to traditional measures of the proximity to Nash equilibria.

**Lemma 3** *(Cai et al. (2022); Golowich et al. (2020a;b)) For a game $\mathcal{G}$ converging to monotone game and a closed convex set $\mathcal{Z}$, $\forall z \in \mathcal{Z}$ and $D > 0$, there exists $P_{\mathcal{G},D}(z) \le D \cdot r_{\mathcal{G}}^{tan}(z)$ and $T_{\mathcal{G},D}(z) \le P_{\mathcal{G},\sqrt{N}D}(z) \le \sqrt{N}D \cdot r_{\mathcal{G}}^{tan}(z)$.*

**Assumptions on the form of time dependence of games.** We introduce the following assumptions which will be applied in lemmas and theorems. The following assumptions are inspired by Benzaid & Lutz (1987); Elaydi & Györi (1995); Elaydi et al. (1999); Feng et al. (2023); Zhao et al. (2020) and are commonly called bounded accumulated perturbations (BAP) assumptions.

**Assumption 2** *$g(z)$ in $\mathcal{G}$ satisfies $\sum_{t=0}^{\infty} \max \|G_t(z)\| < \infty$ where $\mathcal{Z}$ is bounded.*

**Assumption 3** *$G$ is $L_G$-Lipschitz and $\sum_{t=0}^{\infty} \max \|G_t(z)\| < \infty$ and $\sum_{t=0}^{\infty} \|L(G_t)\| < \infty$ where $\mathcal{Z}$ is bounded.*

**Assumption 4** *$G$ is $L_G$-Lipschitz with $G_t(z^*) = 0$ and $\sum_{t=0}^{\infty} \|L(G_t)\| < \infty$.*

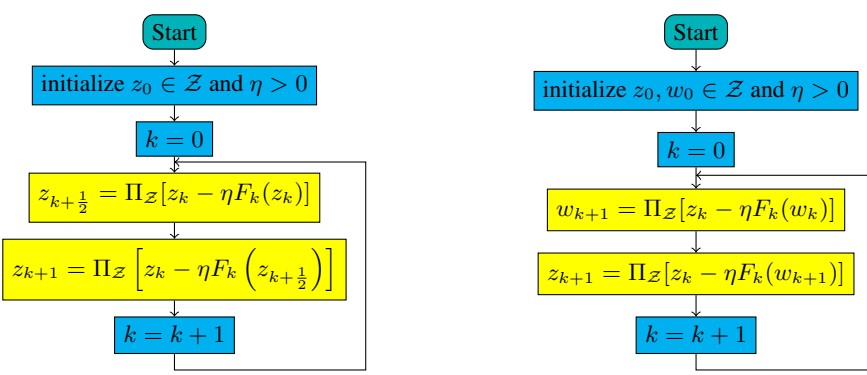

Figure 1: An illustration of the EG (left) and OG (right) algorithms.

Under Assumptions 3 and 4, we also denote $\max L(G_t) = L_G$. For games satisfying Assumption 4, $z_k, w_k$ are bounded with the EG and OG algorithms, and we do not need to assume it. The proof is deferred to Appendices B.2 and C.2. Throughout the paper, each assumption is used independently.

**Remark 1** *In Assumption 4, it is not necessary to assume $\sum_{t=0}^{\infty} \max \|G_t(z)\| < \infty$, because there exists $\sum_{t=0}^{\infty} \max \|G_t(z)\| = \sum_{t=0}^{\infty} \max \|G_t(z) - G_t(z^*)\| \leq \sum_{t=0}^{\infty} \max \|L_G\| \|z - z^*\| \leq \max \|z_t - z^*\| \sum_{t=0}^{\infty} \|L(G_t)\| < \infty$.*

## 3 LAST-ITERATE CONVERGENCE RESULTS OF THE EG ALGORITHM

In this section, we prove that EG with a constant learning rate $\eta$ converges to a Nash equilibrium action at the rate of $O(1/\sqrt{T})$ under any one of Assumptions 2, 3 and 4. The EG algorithm is analyzed in the following steps: firstly, the tangent residual of $\mathcal{G}$ is applied as the measure of proximity in approximating a Nash equilibrium in any iteration; next, the best convergence rate within $T$ steps is shown, which is a small number; finally, it is proved that the square of tangent residual of $\mathcal{G}$ is approximately non-increasing, which means the last-iterate convergence behaviour of the EG algorithm is at least not worse than the best iterate too much and that the last-iterate convergence rate of the EG algorithm is at least not much worse than the convergence rate of infinite series of perturbation.

### 3.1 BEST-ITERATE CONVERGENCE OF THE EG ALGORITHM

To estimate the last-iterate convergence rate of the EG algorithm, we first estimate its best-iterate convergence rate. Lemma 4 shows that there exists $t^* \in [[T]]$ satisfying $\|z_{t^*} - z_{t^*+\frac{1}{2}}\|^2 = O(1/T)$ and $r^{tan}(z_{t^*+1}) = O(1/\sqrt{T})$. The full version and the detailed proof of this lemma are deferred to Appendix B.3.

**Lemma 4** *For a game $\mathcal{G}$ converging to a monotone game with any closed convex set $\mathcal{Z} \subseteq \mathbb{R}^n$ and monotone and $L$-Lipschitz operator $F_{\infty} : \mathcal{Z} \to \mathbb{R}^n$, with the EG algorithm, let $z^*$ be a solution of the game $\mathcal{G}$. Then $\forall T \geq 1$, there exists a $t^* \in [[T]]$ and $C_1, C_2, C_3 > 0$ satisfying*

$$\left\| z_{t^*} - z_{t^*+\frac{1}{2}} \right\|^2 \leq \frac{C_1}{T}$$

*and satisfying*

$$r^{tan}(z_{t^*+1}) \leq \max \left\{ \frac{C_2}{\sqrt{T}} + C_3 \max \|G_{t^*}\| \right\}$$

*under Assumption 2 if $\eta \in \left(0, \frac{1}{L}\right)$; under Assumption 3 if $\eta \in \left(0, \frac{1}{\sqrt{L^2 + 2L_G(L + L_G)}}\right)$;*

$$r^{tan}(z_{t^*+1}) \leq \max \left\{ \frac{C_2}{\sqrt{T}} + L_{G_{t^*}} D \right\}$$

*under Assumption 4 with $D = \max \left\| z^* - z_{t^*+\frac{1}{2}} \right\|$ if $1 - \eta^2 L^2 - 2\eta^2 L_G (L + L_G) - 4\eta L_G > 0$.*

### 3.2 APPROXIMATE MONOTONICITY OF THE EG ALGORITHM

In this subsection, we provide the monotonicity behaviour of the EG algorithm. Since the approach is complicated, the detailed proof of the following theorem is deferred to Appendix B.4.

**Theorem 1** *For a game $\mathcal{G}$ converging to a monotone game, any closed convex set $\mathcal{Z} \subseteq \mathbb{R}^n$ and monotone and $L$-Lipschitz operator $F_{\infty} : \mathcal{Z} \to \mathbb{R}^n$, $\forall \eta \in (0, 1/L)$, $z_k \in \mathcal{Z}$, $r^{tan}(z_k)^2 \geq r^{tan}(z_{k+1})^2 - \max \|G_k\| (4D_{\mathcal{Z}} + 4\eta \max \|F_{\infty}\|)/\eta$. If $\mathcal{Z} = \mathbb{R}^n$, $r^{tan}(z_k)^2 \geq r^{tan}(z_{k+1})^2 - \max \|G_k\| (4D + 2\eta \max \|F_{\infty}\|)/\eta$ where $D = \max\{\|z_k - z_{k+1}\|, \|z_{k+\frac{1}{2}} - z_{k+1}\|\}$, $k \in \mathbb{N}$.*

### 3.3 LAST-ITERATE CONVERGENCE OF THE EG ALGORITHM

This subsection combines the best-iterate results and the approximate monotonicity to estimate the last-iterate convergence rate.

**Theorem 2** (Last-iterate convergence of the EG algorithm) *For a game $\mathcal{G}$ converging to a monotone game, $\forall T \in \mathbb{N}^*, D > 0$,*

$$\max\left\{r^{tan}(z_T), \frac{T_{\mathcal{G},D}(z_T)}{\sqrt{N}D}, \frac{P_{\mathcal{G},D}(z_T)}{D}\right\}$$
$$= O\left(\max\left\{\frac{1}{\sqrt{T}}, \sqrt{\sum_{k=T/2}^{\infty} \max\|G_k\|}, \max\|G_T\|\right\}\right)$$

*where $D = D_{\mathcal{Z}}$ with $\eta \in \left(0, \frac{1}{L}\right)$ under Assumption 2, or $D = D_{\mathcal{Z}}$ with $\eta \in \left(0, \frac{1}{\sqrt{L^2+2L_G(L+L_G)}}\right)$ under Assumption 3, and*

$$\max\left\{r^{tan}(z_T), \frac{T_{\mathcal{G},D}(z_T)}{\sqrt{N}D}, \frac{P_{\mathcal{G},D}(z_T)}{D}\right\}$$
$$= O\left(\max\left\{\frac{1}{\sqrt{T}}, \max_{k \geq T/2} L_{G_k}, \sqrt{\sum_{k=T/2}^{\infty} \max\|G_k\|}\right\}\right)$$

*where $D = \max\|z_k - z_{k+1}\|$, $k \in \mathbb{N}$ with $1 - \eta^2 L^2 - 2\eta^2 L_G(L+L_G) - 4\eta L_G > 0$ under Assumption 4.*

The proof is deferred to Appendix B.5.

## 4 Last-iterate convergence results of the OG algorithm

In this section, we use a method similar to the previous section to show the last-iterate convergence results of the OG algorithm.

We analyze the algorithm with the following steps: firstly, we apply a potential function based on the tangent residual of $\mathcal{G}$, i.e., $\Delta(z_k, w_k) = r^{tan}(z_k)^2 + \|F_\infty(z_k) - F_\infty(w_k)\|^2$ as the measure of proximity in approximating a Nash equilibrium in the current iteration; next, we show the best convergence rate within $T$ steps, which is a small number; finally, we prove that the potential function of $\mathcal{G}$ is approximately non-increasing, so the last-iterate is at least not worse than the best iterate too much, which means that the last-iterate convergence rate is at least not much worse than the convergence rate of infinite series of perturbation.

### 4.1 Best-iterate convergence of the OG algorithm

To estimate the last-iterate convergence rate of the OG algorithm, we first estimate its best-iterate convergence rate. Lemma 5 shows that there exists $t^* \in [[T]]$ satisfying $\eta^2 \Delta(z_{t^*}, w_{t^*}) \leq O(1/T)$. The proof for the best-iterate convergence rate of the OG algorithm is based on Hsieh et al. (2019); Wei et al. (2021). The full version and the proof details of the lemma are deferred to Appendix C.2.

**Lemma 5** *For a game $\mathcal{G}$ converging to a monotone game with any closed convex set $\mathcal{Z} \subseteq \mathbb{R}^n$ and monotone and L-Lipschitz operator $F_\infty : \mathcal{Z} \to \mathbb{R}^n$, let $z^*$ be a solution of the game $\mathcal{G}$. Then $\forall T \geq 1$, if $\eta \in \left(0, \frac{1}{\sqrt{6}L}\right)$, there exists $t^* \in [[T]]$ and $C_1, C_2, C_3, C_4 > 0$ satisfying*

$$\eta^2 \Delta(z_{t^*}, w_{t^*}) \leq \frac{1}{T}\left(C_1\|z_0 - z^*\|^2 + C_2\|w_0 - z_0\|^2 + C_3 E_{mk^2} + C_4 E_{mk}\right)$$

*where*

$$E_{mk} = \sum_{k=0}^{\infty} \max\|G_k\|$$
$$E_{mk^2} = \sum_{k=0}^{\infty} \max\|G_k\|^2$$

*Under Assumption 3, if $\eta \in \left(0, \frac{1}{\sqrt{6L^2+4LL_G+2L_G^2}}\right)$, there exists $t^* \in [[T]]$ and $C_1, C_2, C_3, C_4 > 0$ satisfying*

$$\eta^2 \Delta(z_{t^*}, w_{t^*}) \leq \frac{1}{T}\left(C_1\|z_0 - z^*\|^2 + C_2\|w_0 - z_0\|^2 + C_3 E_{mk} + C_4 E_{mk^2}\right)$$

*while under Assumption 4, if $\eta$ is small, there exists $t^* \in [[T]]$ and $C_1, C_2, D > 0$ satisfying*

$$\eta^2 \Delta(z_{t^*}, w_{t^*}) \leq \frac{1}{T}\left(C_1\|z_0 - z^*\|^2 + C_2\|w_0 - z_0\|^2 + 2\eta^2 D^2 \sum_{k=1}^{T} L_{G_{k-1}}\left(2L + L_{G_{k-1}}\right)\right)$$

## 4.2 Approximate monotonicity of the OG algorithm

This section shows the approximate monotonicity of $\Delta(z_t, w_t)$ for the OG algorithm. Since the approach is complicated, we defer the detailed proof of the following theorem to Appendix C.3.

**Theorem 3** *For a game $\mathcal{G}$ converging to a monotone game with any closed convex set $\mathcal{Z} \subseteq \mathbb{R}^n$, $\forall \eta \in (0, 1/(2L))$, $z_k \in \mathcal{Z}$, $\Delta(z_t, w_t) \geq \Delta(z_{t+1}, w_{t+1}) - (3D_{\mathcal{Z}} + 4\eta \max \|F_\infty\|) \max \|G_k\|/\eta$ if $\mathcal{Z}$ is bounded, while $\Delta(z_t, w_t) \geq \Delta(z_{t+1}, w_{t+1}) - (3D + 2\eta \max \|F_\infty\|) \max \|G_k\|/\eta$ if $\mathcal{Z} = \mathbb{R}^n$, where $D = \max\{\|w_{k+1} - z_{k+1}\|, \|z_{k+1} - z_k\|\}$, $k \in \mathbb{N}$.*

## 4.3 Last-iterate convergence of the OG algorithm

This subsection shows a formal result on the last-iterate convergence behaviour based on the modified tangent residual. The following theorem states that the players' strategies converge to the Nash equilibrium of the game with a rate not slower than the rate of $O(1/\sqrt{T})$ or depending on the convergence rate of perturbation.

**Theorem 4** (Last-iterate convergence of the OG algorithm) *For a game $\mathcal{G}$ converging to a monotone game, $\forall T \in \mathbb{N}^*$,*

$$
\max \left\{ r^{tan}(w_T), \frac{T_{\mathcal{G},D_{\mathcal{Z}}}(w_T)}{\sqrt{N}D_{\mathcal{Z}}}, \frac{P_{\mathcal{G},D_{\mathcal{Z}}}(w_T)}{D_{\mathcal{Z}}} \right\}
$$
$$
= O\left( \max \left\{ \frac{1}{\sqrt{T}}, \sqrt{\sum_{k=T/2}^{\infty} \max \|G_k\|}, \max \|G_T\| \right\} \right)
$$

*under Assumption 2 with $\eta \in \left(0, \frac{1}{\sqrt{6}L}\right)$ or under Assumption 3 with $\eta \in \left(0, \frac{1}{\sqrt{6L^2+4LL_G+2L_G^2}}\right)$, while there exists*

$$
\max \left\{ r^{tan}(w_T), \frac{T_{\mathcal{G},D}(w_T)}{\sqrt{N}D}, \frac{P_{\mathcal{G},D}(w_T)}{D} \right\}
$$
$$
= O\left( \max \left\{ \frac{1}{\sqrt{T}}, \sqrt{\sum_{k=T/2}^{\infty} \max \|G_k\|}, \max \|G_T\|, L_{G_{T-1}} \right\} \right)
$$

*under Assumption 4 with $\eta \in \left(0, \min\left\{\frac{1}{2(L+L_G)}, \frac{1}{4L_G}\right\}\right)$, where $D = \max\{\|w_{k+1} - z_{k+1}\|, \|z_{k+1} - z_k\|\}$, $k \in \mathbb{N}$.*

The proof is deferred to Appendix C.5.

## 5 Experiments for time-varying games

In this section, we provide some numerical examples for Theorem 2 and Theorem 4 proved in Section 3 and Section 4. The numerical examples are based on examples of bilinear games in Feng et al. (2023). By the following examples, we verify Theorem 2 and Theorem 4.

**Example 1.**

$$f_t(z) = f_\infty(z) + g_t(z)$$

$$
= \left[ \begin{bmatrix} z_{t1} - 1 \\ z_{t2} - 1 \end{bmatrix}^T \begin{bmatrix} 2 & 5 \\ 4 & 1 \end{bmatrix} \begin{bmatrix} z_{t3} + 3 \\ z_{t4} + 3 \end{bmatrix} - \begin{bmatrix} z_{t1} - 1 \\ z_{t2} - 1 \end{bmatrix}^T \begin{bmatrix} 2 & 5 \\ 4 & 1 \end{bmatrix} \begin{bmatrix} z_{t3} + 3 \\ z_{t4} + 3 \end{bmatrix} \right] + \begin{bmatrix} -\frac{40\cos\left((t+1)^i z_{t1}\right) + 40\cos\left((t+1)^i z_{t2}\right) + 20\cos\left((t+1)^i z_{t3}\right) + 20\cos\left((t+1)^i z_{t4}\right)}{(t+1)^{2i}} \\ -\frac{2\cos\left((t+1)^i z_{t1}\right) + 3\cos\left((t+1)^i z_{t2}\right) + 5\cos\left((t+1)^i z_{t3}\right) + 6\cos\left((t+1)^i z_{t4}\right)}{(t+1)^{2i}} \end{bmatrix}
$$

s.t. $\|z_t\| = \left\| \begin{bmatrix} z_{t1} & z_{t2} & z_{t3} & z_{t4} \end{bmatrix}^T \right\| \leq 2$

This example is corresponding to Assumption 2, where $\mathcal{Z} = \{z | \|z\| \leq 2\}$ and $\sum_{t=0}^{\infty} \max \|G_t(z)\| \leq \sum_{t=0}^{\infty} \frac{20\sqrt{10}}{(t+1)^i} < \infty$ if $i > 1$.

**Example 2.**

$$f_t(z) = f_\infty(z) + g_t(z) = \begin{bmatrix} \begin{bmatrix} z_{t1} - 1 \\ z_{t2} - 1 \end{bmatrix}^T \begin{bmatrix} 2 & 5 \\ 4 & 1 \end{bmatrix} \begin{bmatrix} z_{t3} + 3 \\ z_{t4} + 3 \end{bmatrix} \\ -\begin{bmatrix} z_{t1} - 1 \\ z_{t2} - 1 \end{bmatrix}^T \begin{bmatrix} 2 & 5 \\ 4 & 1 \end{bmatrix} \begin{bmatrix} z_{t3} + 3 \\ z_{t4} + 3 \end{bmatrix} \end{bmatrix} + \begin{bmatrix} \frac{200(z_{t1}^2 + z_{t2}^2) + 100(z_{t3}^2 + z_{t4}^2)}{(t+1)^i} \\ \frac{50(z_{t1}^2 + z_{t2}^2) + 60(z_{t3}^2 + z_{t4}^2)}{(t+1)^i} \end{bmatrix}$$

s.t. $\|z_t\| = \left\| [z_{t1} \quad z_{t2} \quad z_{t3} \quad z_{t4}]^T \right\| \leq 2$

This example is corresponding to Assumption 3, where $\mathcal{Z} = \{z | \|z\| \leq 2\}$, $\sum_{t=0}^{\infty} \max \|G_t(z)\| \leq \sum_{t=0}^{\infty} \frac{200\sqrt{10}}{(t+1)^i} < \infty$ if $i > 1$ and $\sum_{t=0}^{\infty} L_{G_t} \leq \sum_{t=0}^{\infty} \frac{400\sqrt{10}}{(t+1)^i} < \infty$ if $i > 1$.

**Example 3.**

$$f_t(z) = f_\infty(z) + g_t(z)$$

$$= \begin{bmatrix} \begin{bmatrix} z_{t1} - 1 \\ z_{t2} - 1 \end{bmatrix}^T \begin{bmatrix} 2 & 5 \\ 4 & 1 \end{bmatrix} \begin{bmatrix} z_{t3} + 3 \\ z_{t4} + 3 \end{bmatrix} \\ -\begin{bmatrix} z_{t1} - 1 \\ z_{t2} - 1 \end{bmatrix}^T \begin{bmatrix} 2 & 5 \\ 4 & 1 \end{bmatrix} \begin{bmatrix} z_{t3} + 3 \\ z_{t4} + 3 \end{bmatrix} \end{bmatrix} + \begin{bmatrix} \frac{50(z_{t1}-1)^2 + 50(z_{t2}-1)^2 + 50(z_{t3}+3)^2 + 50(z_{t4}+3)^2}{(t+1)^i} \\ \frac{40(z_{t1}-1)^2 + 30(z_{t2}-1)^2 + 20(z_{t3}+3)^2 + 10(z_{t4}+3)^2}{(t+1)^i} \end{bmatrix}$$

s.t. $z_t = [z_{t1} \quad z_{t2} \quad z_{t3} \quad z_{t4}]^T \in \mathbb{R}^4$

This example is corresponding to Assumption 4, where $\sum_{t=0}^{\infty} L_{G_t} \leq \sum_{t=0}^{\infty} \frac{200}{(t+1)^i} < \infty$ when $i > 1$.

Note that $\forall \epsilon > 0, \exists N > 0$ as the new initial time so that $\forall T > N, L_{G_T} < \epsilon$.

## 5.1 EXPERIMENTS ON THEOREM 2

In this section, the step size $\eta$ is selected as $0.05$ and the initial point of $z_t$ is selected as $z_0 = [0.25 \quad 0.2 \quad 0.1 \quad 0.35]^T$. As in the common practice shown in Feng et al. (2023), the BAP assumptions, i.e., Assumptions 2, 3 and 4, are applied in the three examples above, respectively.

The experimental results are presented in Figure 2, which show that $r^{tan}(z_t)$ converges to 0 under any one of Assumptions 2, 3 and 4, and all three convergence rates of perturbations can decelerate the convergence rate of learning dynamics as expected, thus support the convergence result in Theorem 2.

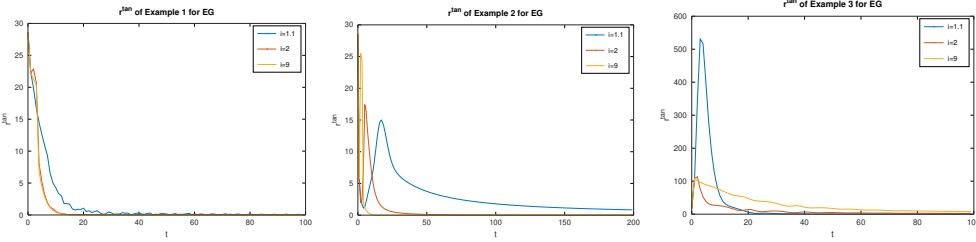

Figure 2: Values of $r^{tan}(z_t)$ for Example 1 (left), Example 2 (middle) and Example 3 (right).

## 5.2 EXPERIMENTS ON THEOREM 4

In this section, the step size $\eta$ is selected as $0.05$ and the initial points of $w_t$ and $z_t$ are selected as $w_0 = z_0 = [0.25 \quad 0.2 \quad 0.1 \quad 0.35]^T$. As in the common practice shown in Feng et al. (2023), the BAP assumptions, i.e., Assumptions 2, 3 and 4, are applied in the three examples above, respectively.

The experimental results are presented in Figure 3, which show that $r^{tan}(w_t)$ converges to 0 under any one of the Assumptions 2, 3 and 4, and all three convergence rates of perturbations can decelerate

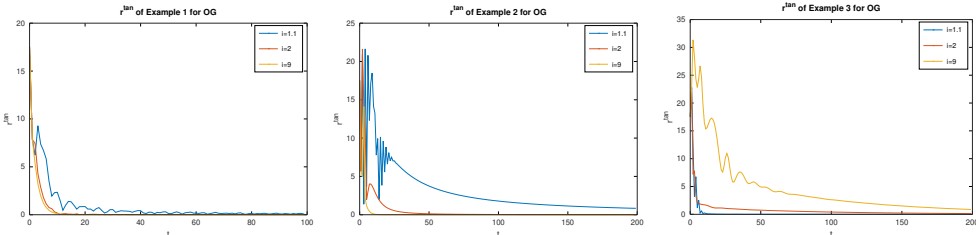

Figure 3: Values of $r^{tan}(w_t)$ for Example 1 (left), Example 2 (middle) and Example 3 (right).

the convergence rate of learning dynamics as expected, thus supporting the convergence result in Theorem 4.

## 6 CONCLUSIONS

In this paper, we provide last-iterate convergence rates of EG and OG algorithms in bounded and some unbounded cases, including unconstrained cases, for time-varying multi-player games converging to monotone games by proving that both algorithms show tight convergence rates compared to related time-invariant games and the property of time-varying function. There exist some interesting future research directions. In our experiments, it is suggested that the convergence rate in bilinear cases can probably be improved. Another interesting direction of investigation is whether and when games vary over time periodically (called "periodic games" in Feng et al. (2023)) or stochastic games show similar results for EG and OG algorithms.

ACKNOWLEDGMENTS

This work was supported by the Natural Science Foundation of China (62276242), National Aviation Science Foundation (2022Z071078001), Dreams Foundation of Jianghuai Advance Technology Center (2023-ZM01Z001).

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

## A  ADDITIONAL AUXILIARY LEMMAS

In this section, we present other lemmas frequently used in appendices required for proving the convergence behaviours of EG and OG. The following lemmas are fundamental to the analysis.

**Lemma 6** *(Lang (1993)) Any continuous function on a closed bounded set within $\mathbb{R}^n$ is bounded.*

**Lemma 7** *(Little et al. (2022)) If $\{a_n\}$ is a sequence of non-negative numbers, then the series $\sum_{j=1}^{\infty} a_j$ and the product $\prod_{j=1}^{\infty}(1 + a_j)$ either both converge or both diverge.*

The following two lemmas state known useful properties of the tangent residual.

**Lemma 8** *(Cai et al. (2022)) Let $\mathcal{Z} \subseteq \mathbb{R}^n$ be a closed convex set and $F : \mathcal{Z} \to \mathbb{R}$ be the operator defined in Section 2. $\forall z \in \mathcal{Z}$, let $c(z) = \Pi_{N(z)}[-F_\infty(z)]$. Then we have*

- $r^{tan}(z) = \|F_\infty(z) + c(z)\|$,

- $\langle F_\infty(z) + c(z), c(z) \rangle = 0$,

- $\langle F_\infty(z) + c(z), a \rangle \geq 0$, $\forall a \in N(z)$.

**Lemma 9** *(Cai et al. (2022)) (Equivalent definitions of the tangent residual) For any closed convex set $\mathcal{Z}$ and operator $F : \mathcal{Z} \to \mathbb{R}^n$, define $N_{\mathcal{Z}}(z)$ as the normal cone of $z$ and $J_{\mathcal{Z}}(z) := \{z\} + T_{\mathcal{Z}}(z)$, where $T_{\mathcal{Z}}(z) = \{z' \in \mathbb{R}^n : \langle z', a \rangle \leq 0, \forall a \in N_{\mathcal{Z}}(z)\}$ is the tangent cone of $z$. Then all of the following quantities are equivalent:*

1. $\sqrt{\|F(z)\|^2 - \max_{a \in \hat{N}_{\mathcal{Z}}(z), \langle F(z), a \rangle \leq 0} \langle F(z), a \rangle^2}$

2. $\min_{a \in \hat{N}_{\mathcal{Z}}(z), \langle F(z), a \rangle \leq 0} \|F(z) - \langle F(z), a \rangle \cdot a\|$

3. $\left\| \Pi_{T_{\mathcal{Z}}(z)}[-F(z)] \right\|$

4. $\left\| \Pi_{J_{\mathcal{Z}}(z)}[z - F(z)] - z \right\|$

5. $\left\| -F(z) - \Pi_{N_{\mathcal{Z}}(z)} \left[ -F(z) \right] \right\|$

6. $\min_{a \in N_{\mathcal{Z}}(z)} \left\| F(z) + a \right\|$

The following lemma states a useful property of the tangent residual.

**Lemma 10** *Let $\mathcal{Z} \subseteq \mathbb{R}^n$ be a closed convex set and $F : \mathcal{Z} \to \mathbb{R}^n$ be the operator defined in Section 2. If $z_1 = \Pi_{\mathcal{Z}}[z_2 - \eta F_{k_{z_3}}(z_3)]$, then we have*

$$r^{tan}(z_1) \leq \left\| \frac{z_2 - z_1}{\eta} + F_\infty(z_1) - F_\infty(z_3) \right\| + \max \left\{ \left\| G_{k_{z_3}} \right\| \right\}$$

*If $G_{k_{z_3}}$ is also $L_{G_{k_{z_3}}}$-Lipschitz and $G_k(z^*) = 0$, then*

$$r^{tan}(z_1) \leq \left\| \frac{z_2 - z_1}{\eta} + F_\infty(z_1) - F_\infty(z_3) \right\| + L_{G_{k_{z_3}}} D$$

*where $D = \max \| z^* - z_3 \|$.*

*Proof.* Due to $z_1 = \Pi_{\mathcal{Z}}[z_2 - \eta F_{k_{z_3}}(z_3)]$, we have $z_2 - \eta F_{k_{z_3}}(z_3) - z_1 \in N_{\mathcal{Z}}(z_1)$. By that equation and item 6 in Lemma 9, there exists

$$
\begin{aligned}
r^{tan}(z_1) = \min_{c \in N_{\mathcal{Z}}(z_1)} \| F_\infty + c \| &\leq \left\| \frac{z_2 - z_1}{\eta} + F_\infty(z_1) - F_{k_{z_3}}(z_3) \right\| \\
&= \left\| \frac{z_2 - z_1}{\eta} + F_\infty(z_1) - F_\infty(z_3) - G_{k_{z_3}}(z_3) \right\| \\
&\leq \left\| \frac{z_2 - z_1}{\eta} + F_\infty(z_1) - F_\infty(z_3) \right\| + \max \left\{ \left\| G_{k_{z_3}} \right\| \right\}
\end{aligned}
$$

If $G_{k_{z_3}}$ is also $L_{G_{k_{z_3}}}$-Lipschitz and $G_k(z^*) = 0$, then there exists $\left\| G_{k_{z_3}}(z^*) - G_{k_{z_3}}(z_3) \right\| \leq L_{G_{k_{z_3}}} \| z^* - z_3 \| \leq L_{G_{k_{z_3}}} D$ so that

$$r^{tan}(z_1) \leq \left\| \frac{z_2 - z_1}{\eta} + F_\infty(z_1) - F_\infty(z_3) \right\| + L_{G_{k_{z_3}}} D$$

$\square$

In the analysis, the natural residual is applied to estimate the tangent residual. Lemma 11 shows that the tangent residual is the upper bound of the natural residual.

**Definition 7** *For any closed convex set $\mathcal{Z}$ and any monotone operator $F : \mathcal{Z} \to \mathbb{R}^n$, the natural residual at $z \in \mathcal{Z}$ is $r^{nat}_{F,\mathcal{Z}}(z) = \| z - \Pi_{\mathcal{Z}}(z - F(z)) \|$.*

**Lemma 11** *(Cai et al. (2022)) For any closed convex set $\mathcal{Z}$, any monotone operator $F : \mathcal{Z} \to \mathbb{R}^n$ and any $z \in \mathcal{Z}$, $r^{nat}_{F,\mathcal{Z}}(z) \leq r^{tan}_{F,\mathcal{Z}}(z)$.*

## B OMITTED PROOFS FOR LAST-ITERATE CONVERGENCE OF THE EG ALGORITHM

In this section, we provide detailed proof for the last-iterate convergence rate of the EG algorithm:

$$z_{k+\frac{1}{2}} = \Pi_{\mathcal{Z}} \left[ z_k - \eta F_k (z_k) \right], z_{k+1} = \Pi_{\mathcal{Z}} \left[ z_k - \eta F_k \left( z_{k+\frac{1}{2}} \right) \right] \tag{3}$$

This section consists of the following parts. First, with a method inspired by Facchinei & Pang (2007) and Korpelevich (1976), the best-iterate convergence rate of $\| z_k - z_{k+\frac{1}{2}} \|$ is proved for the EG algorithm. Then, the upper bound of tangent residual of $z_k$ is proved to be $\max\{O(\| z_k - z_{k+\frac{1}{2}} \|), O(\max \| G_k \|)\}$ if $G_k$ is $L_{G_k}$-Lipschitz and to be $O(\| z_k - z_{k+\frac{1}{2}} \|)$ no matter whether $G_k$ is $L_{G_k}$-Lipschitz. Next, the tangent residual is proved to be either non-increasing or increasing slowly enough across iterates of the EG algorithm. Finally, the last-iterate convergence rate of the EG algorithm is concluded from the conditions above.

## B.1 PREPARATION FOR ANALYZING THE BEST-ITERATE CONVERGENCE BEHAVIOURS OF THE EG ALGORITHM

**Lemma 12** *For a game $\mathcal{G}$ converging to a monotone game with any closed convex set $\mathcal{Z} \subseteq \mathbb{R}^n$, a monotone $L$-Lipschitz operator $F_\infty$ mapping from $\mathcal{Z}$ to $\mathbb{R}^n$ and $z_k \in \mathcal{Z}$, with the EG algorithm, there exists $\left\| z_{k+\frac{1}{2}} - z_{k+1} \right\| \leq \eta(L + L_{G_k}) \left\| z_k - z_{k+\frac{1}{2}} \right\|$ and $\left\| z_k - z_{k+\frac{1}{2}} \right\| \leq \frac{\| z_k - z_{k+1} \|}{1 - \eta L - \eta L_{G_k}}$ when $\eta \in \left( 0, \frac{1}{L + L_G} \right)$ if $G_k$ is $L_{G_k}$-Lipschitz, and there always exists $\left\| z_k - z_{k+\frac{1}{2}} \right\| \leq \frac{\| z_k - z_{k+1} \| + 2\eta \max \| G_k \|}{1 - \eta L}$ when $\eta \in \left( 0, \frac{1}{L} \right)$.*

*Proof.* Since $z_{k+\frac{1}{2}} = \Pi_{\mathcal{Z}}[z_k - \eta F(z_k)]$ and $z_{k+1} = \Pi_{\mathcal{Z}}[z_k - \eta F(z_{k+\frac{1}{2}})]$, due to non-expansiveness of $\Pi_{\mathcal{Z}}$ and $L$-Lipschitzness of $F_\infty$ and $L_{G_t}$-Lipschitzness of $G_t$, there exists

$$
\begin{aligned}
\left\| z_{k+\frac{1}{2}} - z_{k+1} \right\| &\leq \left\| \eta F(z_k) - \eta F\left( z_{k+\frac{1}{2}} \right) \right\| \\
&\leq \left\| \eta F_\infty(z_k) - \eta F_\infty\left( z_{k+\frac{1}{2}} \right) \right\| + \eta \left\| G_k(z_k) - G_k\left( z_{k+\frac{1}{2}} \right) \right\| \\
&\leq \eta L \left\| z_k - z_{k+\frac{1}{2}} \right\| + \eta L_{G_k} \left\| z_k - z_{k+\frac{1}{2}} \right\|
\end{aligned}
$$

$$
\left\| z_{k+\frac{1}{2}} - z_{k+1} \right\| \leq \eta L \left\| z_k - z_{k+\frac{1}{2}} \right\| + 2\eta \max \| G_k \|
$$

Since $\lim_{k \to \infty} L_{G_k} = 0$, there exists

$$
\left\| z_{k+\frac{1}{2}} - z_{k+1} \right\| \leq \eta(L + L_{G_k}) \left\| z_k - z_{k+\frac{1}{2}} \right\|
$$

Hence,

$$
\| z_k - z_{k+1} \| \geq \left\| z_k - z_{k+\frac{1}{2}} \right\| - \left\| z_{k+\frac{1}{2}} - z_{k+1} \right\| \geq \left( 1 - \eta L - \eta L_{G_k} \right) \left\| z_k - z_{k+\frac{1}{2}} \right\|
$$

$$
\| z_k - z_{k+1} \| \geq \left\| z_k - z_{k+\frac{1}{2}} \right\| - \left\| z_{k+\frac{1}{2}} - z_{k+1} \right\| \geq (1 - \eta L) \left\| z_k - z_{k+\frac{1}{2}} \right\| - 2\eta \max \| G_k \|
$$

$\square$

**Lemma 13** *For a game $\mathcal{G}$ converging to a monotone game with any convex set $\mathcal{Z} \subseteq \mathbb{R}^n$, monotone $L$-Lipschitz operator $F_\infty$ mapping from $\mathcal{Z}$ to $\mathbb{R}^n$ and $z^* \in \mathcal{Z}$, there exists*

$$
\| z_{k+1} - z^* \|^2 \leq \| z_k - z^* \|^2 - \left( 1 - \eta^2 L^2 \right) \left\| z_k - z_{k+\frac{1}{2}} \right\|^2 + 4\eta D_{\mathcal{Z}} \max \| G_k \|
$$

*where $D_{\mathcal{Z}} = \max_{x_1, x_2 \in \mathcal{Z}} \| x_1 - x_2 \|$ defined in Section 2. If $G_k$ is also $L_{G_k}$-Lipschitz,*

$$
\begin{aligned}
\| z_{k+1} - z^* \|^2 &\leq \| z_k - z^* \|^2 - \left( 1 - \eta^2 L^2 - 2\eta^2 L_{G_k}(L + L_{G_k}) \right) \left\| z_k - z_{k+\frac{1}{2}} \right\|^2 \\
&\quad + 2\eta \left( \max \left\| G_k\left( z_{k+\frac{1}{2}} \right) \right\| \left\| z_{k+\frac{1}{2}} - z^* \right\| \right)
\end{aligned}
$$

*while under Assumption 4,*

$$
\| z_{k+1} - z^* \|^2 \leq (1 + 4\eta L_{G_k}) \| z_k - z^* \|^2 - \left( 1 - \eta^2 L^2 - 2\eta^2 L_{G_k}(L + L_{G_k}) - 4\eta L_{G_k} \right) \left\| z_k - z_{k+\frac{1}{2}} \right\|^2
$$

*Proof.* Due to Pythagorean equality which is also used in the proof of Lemma 10 in Cai et al. (2022), $\left\langle z_{k+1} - z^*, z_{k+1} - \left( z_k - \eta F\left( z_{k+\frac{1}{2}} \right) \right) \right\rangle \leq 0$. Hence, there exists

$$
\| z_{k+1} - z^* \|^2 \leq \left\| z_k - \eta F_\infty\left( z_{k+\frac{1}{2}} \right) - \eta G_k\left( z_{k+\frac{1}{2}} \right) - z^* \right\|^2 + \left\| z_k - \eta F_\infty\left( z_{k+\frac{1}{2}} \right) - \eta G_k\left( z_{k+\frac{1}{2}} \right) - z_{k+1} \right\|^2
$$

Furthermore, we have

$$
\begin{aligned}
\|z_{k+1} - z^*\|^2 &\leq \left\| z_k - \eta F_\infty\left(z_{k+\frac{1}{2}}\right) - \eta G_k\left(z_{k+\frac{1}{2}}\right) - z^* \right\|^2 - \left\| z_k - \eta F_\infty\left(z_{k+\frac{1}{2}}\right) - \eta G_k\left(z_{k+\frac{1}{2}}\right) - z_{k+1} \right\|^2 \\
&= \|z_k - z^*\|^2 - \|z_k - z_{k+1}\|^2 + 2\eta \left\langle z^* - z_k, F_\infty\left(z_{k+\frac{1}{2}}\right) + G_k\left(z_{k+\frac{1}{2}}\right) \right\rangle \\
&\quad - 2\eta \left\langle z_{k+1} - z_k, F_\infty\left(z_{k+\frac{1}{2}}\right) + G_k\left(z_{k+\frac{1}{2}}\right) \right\rangle \\
&= \|z_k - z^*\|^2 - \|z_k - z_{k+1}\|^2 + 2\eta \left\langle z^* - z_{k+1}, F_\infty\left(z_{k+\frac{1}{2}}\right) + G_k\left(z_{k+\frac{1}{2}}\right) \right\rangle \\
&= \|z_k - z^*\|^2 - \|z_k - z_{k+1}\|^2 + 2\eta \left\langle F_\infty\left(z_{k+\frac{1}{2}}\right), z^* - z_{k+\frac{1}{2}} \right\rangle \\
&\quad + 2\eta \left\langle F_\infty\left(z_{k+\frac{1}{2}}\right), z_{k+\frac{1}{2}} - z_{k+1} \right\rangle + 2\eta \left\langle z^* - z_{k+1}, G_k\left(z_{k+\frac{1}{2}}\right) \right\rangle \\
&\leq \|z_k - z^*\|^2 - \left( \left\| z_k - z_{k+\frac{1}{2}} \right\|^2 + \left\| z_{k+\frac{1}{2}} - z_{k+1} \right\|^2 + 2\left\langle z_k - z_{k+\frac{1}{2}}, z_{k+\frac{1}{2}} - z_{k+1} \right\rangle \right) \\
&\quad + 2\eta \left\langle F_\infty\left(z_{k+\frac{1}{2}}\right), z_{k+\frac{1}{2}} - z_{k+1} \right\rangle + 2\eta \left\langle z^* - z_{k+1}, G_k\left(z_{k+\frac{1}{2}}\right) \right\rangle
\end{aligned}
$$

Since $\left\langle F_\infty\left(z_{k+\frac{1}{2}}\right), z^* - z_{k+\frac{1}{2}} \right\rangle \leq 0$, there exists

$$
\begin{aligned}
\|z_{k+1} - z^*\|^2 &\leq \|z_k - z^*\|^2 - \left\| z_k - z_{k+\frac{1}{2}} \right\|^2 - \left\| z_{k+\frac{1}{2}} - z_{k+1} \right\|^2 + 2\eta \left\langle z^* - z_{k+\frac{1}{2}}, G_k\left(z_{k+\frac{1}{2}}\right) \right\rangle \\
&\quad + 2\eta \left\langle z_{k+\frac{1}{2}} - z_{k+1}, G_k\left(z_{k+\frac{1}{2}}\right) \right\rangle - 2\left\langle z_k - \eta F_\infty\left(z_{k+\frac{1}{2}}\right) - z_{k+\frac{1}{2}}, z_{k+\frac{1}{2}} - z_{k+1} \right\rangle \\
&= \|z_k - z^*\|^2 - \left\| z_k - z_{k+\frac{1}{2}} \right\|^2 - \left\| z_{k+\frac{1}{2}} - z_{k+1} \right\|^2 + 2\eta \left\langle z^* - z_{k+\frac{1}{2}}, G_k\left(z_{k+\frac{1}{2}}\right) \right\rangle \\
&\quad - 2\left\langle z_k - \eta\left(F_\infty\left(z_k\right) + G_k\left(z_k\right)\right) - z_{k+\frac{1}{2}}, z_{k+\frac{1}{2}} - z_{k+1} \right\rangle + 2\eta \left\langle z_{k+\frac{1}{2}} - z_{k+1}, G_k\left(z_{k+\frac{1}{2}}\right) \right\rangle \\
&\quad - 2\eta \left\langle z_{k+\frac{1}{2}} - z_{k+1}, G_k\left(z_k\right) \right\rangle - 2\eta \left\langle z_{k+\frac{1}{2}} - z_{k+1}, F_\infty\left(z_k\right) - F_\infty\left(z_{k+\frac{1}{2}}\right) \right\rangle
\end{aligned}
$$

Due to $\left\langle z_k - \eta\left(F_\infty\left(z_k\right) + G_k\left(z_k\right)\right) - z_{k+\frac{1}{2}}, z_{k+\frac{1}{2}} - z_{k+1} \right\rangle = \left\langle z_k - \eta F\left(z_k\right) - z_{k+\frac{1}{2}}, z_{k+\frac{1}{2}} - z_{k+1} \right\rangle \geq 0$, $z_{k+\frac{1}{2}} = \Pi_{\mathcal{Z}}[z_k - \eta F(z_k)]$ and $z_{k+1} \in \mathcal{Z}$, there exists

$$
\begin{aligned}
\|z_{k+1} - z^*\|^2 &\leq \|z_k - z^*\|^2 - \left\| z_k - z_{k+\frac{1}{2}} \right\|^2 - \left\| z_{k+\frac{1}{2}} - z_{k+1} \right\|^2 + 2\eta \left\langle z^* - z_{k+\frac{1}{2}}, G_k\left(z_{k+\frac{1}{2}}\right) \right\rangle \\
&\quad + 2\eta \left\langle z_{k+\frac{1}{2}} - z_{k+1}, G_k\left(z_{k+\frac{1}{2}}\right) - G_k\left(z_k\right) \right\rangle - 2\eta \left\langle z_{k+\frac{1}{2}} - z_{k+1}, F_\infty\left(z_k\right) - F_\infty\left(z_{k+\frac{1}{2}}\right) \right\rangle \\
&\leq \|z_k - z^*\|^2 - \left\| z_k - z_{k+\frac{1}{2}} \right\|^2 - \left\| z_{k+\frac{1}{2}} - z_{k+1} \right\|^2 + 2\eta \left\langle z^* - z_{k+1}, G_k\left(z_{k+\frac{1}{2}}\right) \right\rangle \\
&\quad - 2\eta \left\langle z_{k+\frac{1}{2}} - z_{k+1}, G_k\left(z_k\right) \right\rangle + 2\eta L \left\| z_{k+\frac{1}{2}} - z_{k+1} \right\| \left\| z_k - z_{k+\frac{1}{2}} \right\| \\
&= \|z_k - z^*\|^2 - \left\| z_k - z_{k+\frac{1}{2}} \right\|^2 - \left( \left\| z_{k+\frac{1}{2}} - z_{k+1} \right\| - \eta L \left\| z_k - z_{k+\frac{1}{2}} \right\| \right)^2 \\
&\quad + \eta^2 L^2 \left\| z_k - z_{k+\frac{1}{2}} \right\|^2 + 4\eta D_{\mathcal{Z}} \max \|G_k\| \\
&\leq \|z_k - z^*\|^2 - \left(1 - \eta^2 L^2\right) \left\| z_k - z_{k+\frac{1}{2}} \right\|^2 + 4\eta D_{\mathcal{Z}} \max \|G_k\|
\end{aligned}
$$

If $G_k$ is $L_{G_k}$-Lipschitz, according to Lemma 12 , there exists

$$
\begin{aligned}
\|z_{k+1} - z^*\|^2 &\leq \|z_k - z^*\|^2 - \left\|z_k - z_{k+\frac{1}{2}}\right\|^2 - \left\|z_{k+\frac{1}{2}} - z_{k+1}\right\|^2 + 2\eta \left\langle z^* - z_{k+\frac{1}{2}}, G_k\left(z_{k+\frac{1}{2}}\right)\right\rangle \\
&\quad + 2\eta \left\langle z_{k+\frac{1}{2}} - z_{k+1}, G_k\left(z_{k+\frac{1}{2}}\right) - G_k\left(z_k\right)\right\rangle - 2\eta \left\langle z_{k+\frac{1}{2}} - z_{k+1}, F_\infty\left(z_k\right) - F_\infty\left(z_{k+\frac{1}{2}}\right)\right\rangle \\
&\leq \|z_k - z^*\|^2 - \left\|z_k - z_{k+\frac{1}{2}}\right\|^2 - \left\|z_{k+\frac{1}{2}} - z_{k+1}\right\|^2 + 2\eta \left\langle z^* - z_{k+\frac{1}{2}}, G_k\left(z_{k+\frac{1}{2}}\right)\right\rangle \\
&\quad + 2\eta L_{G_k}\left\|z_{k+\frac{1}{2}} - z_{k+1}\right\|\left\|z_{k+\frac{1}{2}} - z_k\right\| + 2\eta L\left\|z_{k+\frac{1}{2}} - z_{k+1}\right\|\left\|z_k - z_{k+\frac{1}{2}}\right\| \\
&= \|z_k - z^*\|^2 - \left\|z_k - z_{k+\frac{1}{2}}\right\|^2 - \left(\left\|z_{k+\frac{1}{2}} - z_{k+1}\right\| - \eta L\left\|z_k - z_{k+\frac{1}{2}}\right\|\right)^2 \\
&\quad + \eta^2 L^2 \left\|z_k - z_{k+\frac{1}{2}}\right\|^2 + 2\eta \left\langle z^* - z_{k+\frac{1}{2}}, G_k\left(z_{k+\frac{1}{2}}\right)\right\rangle \\
&\quad + 2\eta L_{G_k}\left\|z_{k+\frac{1}{2}} - z_{k+1}\right\|\left\|z_{k+\frac{1}{2}} - z_k\right\| \\
&\leq \|z_k - z^*\|^2 - \left(1 - \eta^2 L^2\right)\left\|z_k - z_{k+\frac{1}{2}}\right\|^2 + 2\eta \left\langle z^* - z_{k+\frac{1}{2}}, G_k\left(z_{k+\frac{1}{2}}\right)\right\rangle \\
&\quad + 2\eta L_{G_k}\left\|z_{k+\frac{1}{2}} - z_{k+1}\right\|\left\|z_{k+\frac{1}{2}} - z_k\right\| \\
&\leq \|z_k - z^*\|^2 - \left(1 - \eta^2 L^2\right)\left\|z_k - z_{k+\frac{1}{2}}\right\|^2 + 2\eta \left(\max\left\|G_k\left(z_{k+\frac{1}{2}}\right)\right\|\left\|z_{k+\frac{1}{2}} - z^*\right\|\right) \\
&\quad + 2\eta^2 L_{G_k}\left(L + L_{G_k}\right)\left\|z_k - z_{k+\frac{1}{2}}\right\|^2
\end{aligned}
$$

Under Assumption 4,

$$
\begin{aligned}
\|z_{k+1} - z^*\|^2 &\leq \|z_k - z^*\|^2 - \left(1 - \eta^2 L^2\right)\left\|z_k - z_{k+\frac{1}{2}}\right\|^2 + 2\eta \left\langle z^* - z_{k+\frac{1}{2}}, G_k\left(z_{k+\frac{1}{2}}\right) - G_k\left(z^*\right)\right\rangle \\
&\quad + 2\eta^2 L_{G_k}\left(L + L_{G_k}\right)\left\|z_k - z_{k+\frac{1}{2}}\right\|^2 \\
&\leq \|z_k - z^*\|^2 - \left(1 - \eta^2 L^2\right)\left\|z_k - z_{k+\frac{1}{2}}\right\|^2 + 2\eta L_{G_k}\left\|z_{k+\frac{1}{2}} - z^*\right\|^2 \\
&\quad + 2\eta^2 L_{G_k}\left(L + L_{G_k}\right)\left\|z_k - z_{k+\frac{1}{2}}\right\|^2 \\
&\leq \left(1 + 4\eta L_{G_k}\right)\|z_k - z^*\|^2 - \left(1 - \eta^2 L^2\right)\left\|z_k - z_{k+\frac{1}{2}}\right\|^2 \\
&\quad + \left(2\eta^2 L_{G_k}\left(L + L_{G_k}\right) + 4\eta L_{G_k}\right)\left\|z_k - z_{k+\frac{1}{2}}\right\|^2
\end{aligned}
$$

$\square$

## B.2 BOUNDEDNESS OF THE EG ALGORITHM IN $(L + L_{G_k})$-LIPSCHITZ GAMES

This section shows that $z_k$ across iterates of the EG algorithm are bounded under Assumption 4.

**Lemma 14** *For a game $\mathcal{G}$ converging to a monotone game with any closed convex set $\mathcal{Z} \subseteq \mathbb{R}^n$, monotone $L$-Lipschitz operator $F_\infty$ mapping from $\mathcal{Z}$ to $\mathbb{R}^n$ and $z_k \in \mathcal{Z}$, with the EG algorithm under Assumption 4, there exists $C \in \mathbb{R}$ so that $\|z_k - z^*\| \leq C$.*

*Proof.* Due to Assumption 4, $\lim_{k \to \infty} G_k = 0$. Hence, there exists a $N \in \mathbb{R}$ so that $\forall k > N$, $1 - \eta^2 L^2 - 2\eta^2 L_{G_k}\left(L + L_{G_k}\right) - 4\eta L_{G_k} > 0$, and then

$$
\|z_{k+1} - z^*\|^2 \leq \left(1 + 4\eta L_{G_k}\right)\|z_k - z^*\|^2
$$

By the equation above, there exists

$$
\begin{aligned}
\|z_{T+1} - z^*\|^2 &\leq \|z_N - z^*\|^2 \prod_{k=N}^{T}\left(1 + 4\eta L_{G_k}\right) \\
&\leq \|z_N - z^*\|^2 \prod_{k=N}^{\infty}\left(1 + 4\eta L_{G_k}\right)
\end{aligned}
$$

Therefore, $C \geq \|z_k - z^*\|$ when $C = \max \left\{ \|z_N - z^*\|^2 \prod_{k=N}^{\infty} (1 + 4\eta L_{G_k}), \|z_0 - z^*\|^2, \right.$
$\left. \|z_1 - z^*\|^2, \cdots, \|z_N - z^*\|^2 \right\}$. $\qquad \square$

## B.3 BEST-ITERATE CONVERGENCE OF THE TANGENT RESIDUAL OF THE EG ALGORITHM

**Lemma 15** *For a game $\mathcal{G}$ converging to a monotone game with any convex set $\mathcal{Z} \subseteq \mathbb{R}^n$, monotone $L$-Lipschitz operator $F_\infty$ mapping from $\mathcal{Z}$ to $\mathbb{R}^n$ and $z^* \in \mathcal{Z}$, there exists*

$$r^{tan}(z_{k+1}) \leq \left( \frac{1 + \eta L + (\eta L)^2}{\eta} \right) \left\| z_k - z_{k+\frac{1}{2}} \right\| + (3 + 2\eta L) \max \{ \|G_k\| \}$$

*If $G_k$ is $L_{G_k}$-Lipschitz and $G_t(z^*) = 0$, there exists*

$$r^{tan}(z_{k+1}) \leq \left( \frac{1 + \eta L + (\eta L)^2}{\eta} + (1 + \eta L) L_G \right) \left\| z_k - z_{k+\frac{1}{2}} \right\| + L_{G_k} D$$

*where $D = \max \left\| z^* - z_{k+\frac{1}{2}} \right\|$.*

*Proof.* Due to Lemma 10, there exists

$$r^{tan}(z_{k+1}) \leq \left\| \frac{z_k - z_{k+1}}{\eta} + F_\infty(z_{k+1}) - F_\infty \left( z_{k+\frac{1}{2}} \right) \right\| + \max \{ \|G_k\| \}$$

Hence,

$$\begin{aligned} r^{tan}(z_{k+1}) &\leq \left\| \frac{z_k - z_{k+\frac{1}{2}}}{\eta} \right\| + \left\| \frac{z_{k+1} - z_{k+\frac{1}{2}}}{\eta} \right\| + L \left\| z_{k+1} - z_{k+\frac{1}{2}} \right\| + \max \{ \|G_k\| \} \\ &\leq \left( \frac{1}{\eta} + \left( \frac{1 + \eta L}{\eta} \right) \eta L \right) \left\| z_k - z_{k+\frac{1}{2}} \right\| + \left( 2\eta \left( \frac{1 + \eta L}{\eta} \right) + 1 \right) \max \{ \|G_k\| \} \\ &= \left( \frac{1 + \eta L + (\eta L)^2}{\eta} \right) \left\| z_k - z_{k+\frac{1}{2}} \right\| + (3 + 2\eta L) \max \{ \|G_k\| \} \end{aligned}$$

If $G_k$ is $L_{G_k}$-Lipschitz and $G_t(z^*) = 0$, there exists

$$r^{tan}(z_{k+1}) \leq \left\| \frac{z_k - z_{k+1}}{\eta} + F_\infty(z_{k+1}) - F_\infty \left( z_{k+\frac{1}{2}} \right) \right\| + L_{G_k} D$$

Hence,

$$\begin{aligned} r^{tan}(z_{k+1}) &\leq \left\| \frac{z_k - z_{k+1}}{\eta} \right\| + \left\| F_\infty(z_{k+1}) - F_\infty \left( z_{k+\frac{1}{2}} \right) \right\| + L_{G_k} D \\ &\leq \left\| \frac{z_k - z_{k+\frac{1}{2}}}{\eta} \right\| + \left\| \frac{z_{k+1} - z_{k+\frac{1}{2}}}{\eta} \right\| + L \left\| z_{k+1} - z_{k+\frac{1}{2}} \right\| + L_{G_k} D \\ &\leq \left( \frac{1}{\eta} + L + L_G + \eta L (L + L_G) \right) \left\| z_k - z_{k+\frac{1}{2}} \right\| + L_{G_k} D \\ &= \left( \frac{1 + \eta L + (\eta L)^2}{\eta} + (1 + \eta L) L_G \right) \left\| z_k - z_{k+\frac{1}{2}} \right\| + L_{G_k} D \end{aligned}$$

$\qquad \square$

**Full version of Lemma 4.** *For a game $\mathcal{G}$ converging to a monotone game with any closed convex set $\mathcal{Z} \subseteq \mathbb{R}^n$ and monotone and $L$-Lipschitz operator $F_\infty : \mathcal{Z} \to \mathbb{R}^n$, with the EG algorithm, let $z^*$ be a solution of the game $\mathcal{G}$. Then $\forall T \geq 1$, there exists a $t^* \in [[T]]$ satisfying*

$$\left\| z_{t^*} - z_{t^*+\frac{1}{2}} \right\|^2 \leq \frac{C_1}{T}$$

*and satisfying*

$$r^{tan}(z_{t^*+1}) \leq \max \left\{ \frac{C_2}{\sqrt{T}} + C_3 \max \|G_{t^*}\| \right\}$$

*under Assumptions 2 and 3,*

$$r^{tan}(z_{t^*+1}) \leq \max\left\{\frac{C_2}{\sqrt{T}} + L_{G_{t^*}}D\right\}$$

*under Assumption 4, where*

$$C_1 = \frac{\|z_0 - z^*\|^2 + 4\eta D_{\mathcal{Z}}\sum_{k=0}^{\infty}(\max\|G_k\|)}{1 - \eta^2 L^2}$$

$$C_2 = \frac{1 + \eta L + (\eta L)^2}{\eta}\sqrt{\frac{\|z_0 - z^*\|^2 + 4\eta D_{\mathcal{Z}}\sum_{k=0}^{\infty}(\max\|G_k\|)}{1 - \eta^2 L^2}}$$

$$C_3 = 3 + 2\eta L$$

*under Assumption 2 if $\eta \in \left(0, \frac{1}{L}\right)$;*

$$C_1 = \frac{\|z_0 - z^*\|^2 + 2\eta D_{\mathcal{Z}}\sum_{k=0}^{\infty}\max\|G_k\|}{1 - \eta^2 L^2 - 2\eta^2 L_G(L + L_G)}$$

$$C_2 = \frac{1 + \eta L + (\eta L)^2}{\eta}\sqrt{\frac{\|z_0 - z^*\|^2 + 2\eta D_{\mathcal{Z}}\sum_{k=0}^{\infty}\max\|G_k\|}{1 - \eta^2 L^2 - 2\eta^2 L_G(L + L_G)}}$$

$$C_3 = 3 + 2\eta L$$

*under Assumption 3 if $\eta \in \left(0, \frac{1}{\sqrt{L^2 + 2L_G(L+L_G)}}\right)$;*

$$C_1 = \frac{\|z_0 - z^*\|^2 \prod_{k=0}^{\infty}(1 + 4\eta L_{G_k})}{1 - \eta^2 L^2 - 2\eta^2 L_G(L + L_G) - 4\eta L_G}$$

$$C_2 = \left(\frac{1 + \eta L + (\eta L)^2}{\eta} + (1 + \eta L)L_G\right)\sqrt{\frac{\|z_0 - z^*\|^2 \prod_{k=0}^{\infty}(1 + 4\eta L_{G_k})}{1 - \eta^2 L^2 - 2\eta^2 L_G(L + L_G) - 4\eta L_G}}$$

*under Assumption 4 with $D = \max\left\|z^* - z_{t^*+\frac{1}{2}}\right\|$ if $1 - \eta^2 L^2 - 2\eta^2 L_G(L + L_G) - 4\eta L_G > 0$.*

*Proof.* Due to Lemma 13, there exists

$$\|z_0 - z^*\|^2 \geq \|z_{T+1} - z^*\|^2 + \left(1 - \eta^2 L^2\right)\sum_{k=0}^{T}\left\|z_k - z_{k+\frac{1}{2}}\right\|^2 - \sum_{k=0}^{T} 4\eta\max\|G_k D_{\mathcal{Z}}\|$$

$$\geq \left(1 - \eta^2 L^2\right)\sum_{k=0}^{T}\left\|z_k - z_{k+\frac{1}{2}}\right\|^2 - \sum_{k=0}^{T} 4\eta\max\|G_k D_{\mathcal{Z}}\|$$

$$\geq \left(1 - \eta^2 L^2\right)\sum_{k=0}^{T}\left\|z_k - z_{k+\frac{1}{2}}\right\|^2 - 4\eta D_{\mathcal{Z}}\sum_{k=0}^{\infty}\max\|G_k\|$$

Hence,

$$\left\|z_{t^*} - z_{t^*+\frac{1}{2}}\right\|^2 \leq \frac{1}{T(1 - \eta^2 L^2)}\left(\|z_0 - z^*\|^2 + 4\eta D_{\mathcal{Z}}\sum_{k=0}^{\infty}(\max\|G_k\|)\right)$$

Due to Lemma 15,

$$r^{tan}(z_{t^*+1}) \leq \sqrt{\frac{\|z_0 - z^*\|^2 + 4\eta D_{\mathcal{Z}}\sum_{k=0}^{\infty}(\max\|G_k\|)}{T(1 - \eta^2 L^2)}\left(\frac{1 + \eta L + (\eta L)^2}{\eta}\right)^2} + (3 + 2\eta L)\max\{\|G_{t^*}\|\}$$

If $G_k$ is $L_{G_k}$-Lipschitz, there exists

$$\|z_0 - z^*\|^2 \geq \|z_{T+1} - z^*\|^2 + \sum_{k=0}^{T}\left(1 - \eta^2 L^2 - 2\eta^2 L_{G_k}(L + L_{G_k})\right)\left\|z_k - z_{k+\frac{1}{2}}\right\|^2$$

$$- \sum_{k=0}^{T}\left(2\eta\max\|G_k\|D_{\mathcal{Z}}\right)$$

$$\geq \sum_{k=0}^{T}\left(1 - \eta^2 L^2 - 2\eta^2 L_{G_k}(L + L_{G_k})\right)\left\|z_k - z_{k+\frac{1}{2}}\right\|^2 - \sum_{k=0}^{T}\left(2\eta\max\|G_k\|D_{\mathcal{Z}}\right)$$

For $N \in \mathbb{N}^*$, when $\forall k > N, 1 - \eta^2 L^2 - 2\eta^2 L_{G_k} (L + L_{G_k}) > 0$, there exists

$$\|z_0 - z^*\|^2 \geq \sum_{k=0}^T \left(1 - \eta^2 L^2 - 2\eta^2 L_G (L + L_G)\right) \left\|z_k - z_{k+\frac{1}{2}}\right\|^2 - \sum_{k=0}^T \left(2\eta \max \|G_k\| D_{\mathcal{Z}}\right)$$

Hence,

$$\left\|z_{t^*} - z_{t^*+\frac{1}{2}}\right\|^2 \leq \frac{1}{T(1 - \eta^2 L^2 - 2\eta^2 L_G (L + L_G))} \left(\|z_0 - z^*\|^2 + 2\eta D_{\mathcal{Z}} \sum_{k=0}^\infty \max \|G_k\|\right)$$

Due to Lemma 15,

$$r^{tan}(z_{t^*+1}) \leq \sqrt{\frac{\|z_0 - z^*\|^2 + 2\eta D_{\mathcal{Z}} \sum_{k=0}^\infty \max \|G_k\|}{T(1 - \eta^2 L^2 - 2\eta^2 L_G (L + L_G))}} \left(\frac{1 + \eta L + (\eta L)^2}{\eta}\right)^2 + (3 + 2\eta L) \max \{\|G_{t^*}\|\}$$

Under Assumption 4,

$$\|z_{T+1} - z^*\|^2 \leq \prod_{k=0}^T \left(1 + 4\eta L_{G_k}\right) \|z_0 - z^*\|^2$$
$$- \sum_{k=0}^{T-1} \left(1 - \eta^2 L^2 - 2\eta^2 L_{G_k} (L + L_{G_k}) - 4\eta L_{G_k}\right) \prod_{i=k+1}^T (1 + 4\eta L_{G_i}) \left\|z_k - z_{k+\frac{1}{2}}\right\|^2$$

For $N \in \mathbb{N}^*$, when $\forall k > N, 1 - \eta^2 L^2 - 2\eta^2 L_G (L + L_G) - 4\eta L_G > 0$, there exists

$$\|z_0 - z^*\|^2 \geq \frac{\sum_{k=0}^{T-1} \left(1 - \eta^2 L^2 - 2\eta^2 L_G (L + L_G) - 4\eta L_G\right) \prod_{i=k+1}^T (1 + 4\eta L_{G_i})}{\prod_{k=0}^T (1 + 4\eta L_{G_k})} \left\|z_{t^*} - z_{t^*+\frac{1}{2}}\right\|^2$$

Then, since $\prod_{k=0}^T (1 + 4\eta L_{G_k}) \leq \prod_{k=0}^\infty (1 + 4\eta L_{G_k})$, there exists

$$\left\|z_{t^*} - z_{t^*+\frac{1}{2}}\right\|^2 \leq \frac{\|z_0 - z^*\|^2 \prod_{k=0}^\infty (1 + 4\eta L_{G_k})}{T \left(1 - \eta^2 L^2 - 2\eta^2 L_G (L + L_G) - 4\eta L_G\right)}$$

Due to Lemma 15,

$$r^{tan}(z_{t^*+1}) \leq \sqrt{\frac{\|z_0 - z^*\|^2 \prod_{k=0}^\infty (1 + 4\eta L_{G_k})}{T \left(1 - \eta^2 L^2 - 2\eta^2 L_G (L + L_G) - 4\eta L_G\right)}} \left(\frac{1 + \eta L + (\eta L)^2}{\eta} + (1 + \eta L) L_G\right) + L_{G_{t^*}} D$$

$\qquad\qquad\qquad\qquad\qquad\qquad\qquad\qquad\qquad\qquad\qquad\qquad\qquad\qquad\qquad\qquad\qquad\qquad\qquad\square$

## B.4 Last-iterate convergence of the EG algorithm with a constant step size

**Restatement of Theorem 1.** *For a game $\mathcal{G}$ converging to a monotone game, $\forall \eta \in \left(0, \frac{1}{L}\right)$, $z_k \in \mathcal{Z}$,*

$$r^{tan}(z_k)^2 \geq r^{tan}(z_{k+1})^2 - \max \|G_k\| \left(\frac{4D_{\mathcal{Z}} + 4\eta \max \|F_\infty\|}{\eta}\right)$$

*If $\mathcal{Z} = \mathbb{R}^n$, there exists*

$$r^{tan}(z_k)^2 \geq r^{tan}(z_{k+1})^2 - \max \|G_k\| \left(\frac{4D + 2\eta \max \|F_\infty\|}{\eta}\right)$$

*where $D = \max\{\|z_k - z_{k+1}\|, \|z_{k+\frac{1}{2}} - z_{k+1}\|\}$, $k \in \mathbb{N}$.*

*Proof.* In this proof, "LHS" stands for "left-hand side". Let $c_k = \Pi_{N_{\mathcal{Z}}(z_k)}(-F_\infty(z_k))$ and $c_{k+1} = \Pi_{N_{\mathcal{Z}}(z_{k+1})}(-F_\infty(z_{k+1}))$. By Lemma 8, there exists

$$\eta^2 r^{tan}(z_k)^2 - \eta^2 r^{tan}(z_{k+1})^2 = \|\eta F_\infty(z_k) + \eta c_k\|^2 - \|\eta F_\infty(z_{k+1}) + \eta c_{k+1}\|^2 \qquad (4)$$

Considering that $F_\infty$ is both monotone and $L$-Lipschitz and $L < \frac{1}{\eta}$, there exists

$$-\left(\left\|z_{k+\frac{1}{2}} - z_{k+1}\right\|^2 - \left\|\eta F_\infty\left(z_{k+\frac{1}{2}}\right) - \eta F_\infty(z_{k+1})\right\|^2\right) \leq 0 \qquad (5)$$

$$-2 \left\langle \eta F_\infty(z_{k+1}) - \eta F_\infty(z_k), z_{k+1} - z_k \right\rangle \leq 0 \qquad (6)$$

Since $z_{k+\frac{1}{2}} = \Pi_{\mathcal{Z}}(z_k - \eta F(z_k))$ and $z_{k+1} = \Pi_{\mathcal{Z}}\left(z_k - \eta F\left(z_{k+\frac{1}{2}}\right)\right)$, there exists $z_k - \eta F_k(z_k) - z_{k+\frac{1}{2}} \in N\left(z_{k+\frac{1}{2}}\right)$ and $z_k - \eta F_k\left(z_{k+\frac{1}{2}}\right) - z_{k+1} \in N(z_{k+1})$. Hence,

$$-2\left\langle z_k - \eta F_k(z_k) - z_{k+\frac{1}{2}}, z_{k+\frac{1}{2}} - z_{k+1} \right\rangle \leq 0 \tag{7}$$

$$-2\left\langle z_k - \eta F_k\left(z_{k+\frac{1}{2}}\right) - z_{k+1}, z_{k+1} - z_k \right\rangle \leq 0 \tag{8}$$

$$-2\left\langle \eta c_k, z_k - z_{k+\frac{1}{2}} \right\rangle \leq 0 \tag{9}$$

By Lemma 8, there exists

$$-2\left\langle \eta c_{k+1} + \eta F_\infty(z_{k+1}), z_k - \eta F_k\left(z_{k+\frac{1}{2}}\right) - z_{k+1} \right\rangle \leq 0 \tag{10}$$

$$-2\left\langle \eta c_{k+1} + \eta F_\infty(z_{k+1}), -\eta c_{k+1} \right\rangle = 0 \tag{11}$$

Hence, there exists

Expression (4) + LHS of Inequality (5) + LHS of Inequality (6) + LHS of Inequality (7)
+LHS of Inequality (8) + LHS of Inequality (9) + LHS of Inequality (10) + LHS of Equation (11)

$$= \left\| \eta F_\infty(z_k) + \eta c_k - z_k + z_{k+\frac{1}{2}} \right\|^2 + \left\| \eta F_\infty\left(z_{k+\frac{1}{2}}\right) + \eta c_{k+1} - z_k + z_{k+1} \right\|^2 + 2\eta\left\langle G_k(z_k), z_{k+\frac{1}{2}} - z_{k+1} \right\rangle$$

$$+2\eta\left\langle G_k\left(z_{k+\frac{1}{2}}\right), z_{k+1} - z_k \right\rangle + 2\eta\left\langle G_k\left(z_{k+\frac{1}{2}}\right), \eta c_{k+1} + \eta F_\infty(z_{k+1}) \right\rangle$$

$$\geq 2\eta\left\langle G_k(z_k), z_{k+\frac{1}{2}} - z_{k+1} \right\rangle + 2\eta\left\langle G_k\left(z_{k+\frac{1}{2}}\right), z_{k+1} - z_k \right\rangle + 2\eta\left\langle G_k\left(z_{k+\frac{1}{2}}\right), \eta c_{k+1} + \eta F_\infty(z_{k+1}) \right\rangle$$

$$\geq -4\eta\left(D_{\mathcal{Z}}\max\|G_k\|\right) - 2\eta^2\left(2\max\|F_\infty\|\max\|G_k\|\right) \tag{12}$$

which means

$$r^{tan}(z_k)^2 \geq r^{tan}(z_{k+1})^2 - \max\|G_k\|\left(\frac{4D_{\mathcal{Z}} + 4\eta\max\|F_\infty\|}{\eta}\right) \tag{13}$$

If $\mathcal{Z} = \mathbb{R}^n$, $\eta c_{k+1} = 0$. By Lemma 14, there exists

Expression (4) + LHS of Inequality (5) + LHS of Inequality (6) + LHS of Inequality (7)
+LHS of Inequality (8) + LHS of Inequality (9) + LHS of Inequality (10) + LHS of Equation (11)

$$\geq 2\eta\left\langle G_k(z_k), z_{k+\frac{1}{2}} - z_{k+1} \right\rangle + 2\eta\left\langle G_k\left(z_{k+\frac{1}{2}}\right), z_{k+1} - z_k \right\rangle + 2\eta\left\langle G_k\left(z_{k+\frac{1}{2}}\right), \eta c_{k+1} + \eta F_\infty(z_{k+1}) \right\rangle$$

$$\geq -4\eta(D\max\|G_k\|) - 2\eta^2\left(\max\|F_\infty\|\max\|G_k\|\right)$$

which means

$$r^{tan}(z_k)^2 \geq r^{tan}(z_{k+1})^2 - \max\|G_k\|\left(\frac{4D + 2\eta\max\|F_\infty\|}{\eta}\right) \tag{14}$$

$\square$

Now we combine all the results above in this section in Theorem 5 for the proof of Theorem 2.

**Theorem 5** *For a game $\mathcal{G}$ converging to a monotone game, $\forall T = T_1 + T_2, T_1, T_2 \in \mathbb{N}^*, D_{\mathcal{Z}} > 0$, the convergence rate of the EG algorithm satisfies*

$$r^{tan}\left(z_{T_1+T_2+1}\right) \leq \sqrt{\left(\frac{C_2}{\sqrt{T_2}} + C_3 \max_{k\in[T_1, T_1+T_2]}\|G_k\|\right)^2 + \frac{4D_{\mathcal{Z}} + 4\eta\max\|F_\infty\|}{\eta}\sum_{k=T_1}^{\infty}\max\|G_k\|}$$

*when $G$ satisfies Assumption 2 if $\eta \in \left(0, \frac{1}{L}\right)$ or $G$ satisfies Assumption 3 if $\eta \in \left(0, \frac{1}{\sqrt{L^2 + 2L_G(L+L_G)}}\right)$, and*

$$r^{tan}\left(z_{T_1+T_2+1}\right) \leq \sqrt{\left(\frac{C_2}{\sqrt{T_2}} + \max_{t\in[T_1, T_1+T_2]}L_{G_k}D_0\right)^2 + \frac{4D + a\eta\max\|F_\infty\|}{\eta}\sum_{k=T_1}^{\infty}\max\|G_k\|}$$

*when $G$ satisfies Assumption 4 with $D_0 = \max \left\| z^* - z_{t^*+\frac{1}{2}} \right\|$ and $D = \max\{\|z_k - z_{k+1}\|, \|z_{k+\frac{1}{2}} - z_{k+1}\|\}$, $k \in \mathbb{N}$ if $1 - \eta^2 L^2 - 2\eta^2 L_G (L + L_G) - 4\eta L_G > 0$, and for unconstrained cases $a = 2$ and for constrained cases $a = 4$. Under Assumption 2,*

$$C_2 = \frac{1 + \eta L + (\eta L)^2}{\eta} \sqrt{\frac{\max_{t \in \mathbb{N}} \|z_t - z^*\|^2 + 4\eta D_{\mathcal{Z}} \sum_{k=0}^{\infty} (\max \|G_k\|)}{1 - \eta^2 L^2}}$$

$$C_3 = 3 + 2\eta L$$

*Under Assumption 3,*

$$C_2 = \left( \frac{1 + \eta L + (\eta L)^2}{\eta} + (1 + \eta L)L_G \right) \sqrt{\frac{\max_{t \in \mathbb{N}} \|z_t - z^*\|^2 + 2\eta D_{\mathcal{Z}} \sum_{k=0}^{\infty} \max \|G_k\|}{1 - \eta^2 L^2 - 2\eta^2 L_G (L + L_G)}}$$

$$C_3 = 3 + 2\eta L$$

*Under Assumption 4,*

$$C_2 = \left( \frac{1 + \eta L + (\eta L)^2}{\eta} + (1 + \eta L)L_G \right) \sqrt{\frac{\max_{t \in \mathbb{N}} \|z_t - z^*\|^2 \prod_{k=0}^{\infty} (1 + 4\eta L_{G_k})}{1 - \eta^2 L^2 - 2\eta^2 L_G (L + L_G) - 4\eta L_G}}$$

*Proof.* Due to Lemmas 13, 14 and 15, there exists $t^* \in (T_1, T_1 + T_2] \cap \mathbb{N}^*$ so that $r^{tan}(z_{t^*+1})^2 = O(1/T_2)$. Since $T_1 < t^* + 1$, according to Theorem 1, there exists

$$r^{tan}(z_{T_1+T_2+1})^2 \leq r^{tan}(z_{t^*+1})^2 + \left( \frac{4D_{\mathcal{Z}} + 4\eta \max \|F_\infty\|}{\eta} \right) \sum_{k=T_1}^{\infty} \max \|G_k\|$$

under any one of Assumptions 2 and 3, while there exists

$$r^{tan}(z_{T_1+T_2+1})^2 \leq r^{tan}(z_{t^*+1})^2 + \frac{4D + a\eta \max \|F_\infty\|}{\eta} \sum_{k=T_1}^{\infty} \max \|G_k\|$$

under Assumption 4 with unconstrained games and $a = 2$. Considering that all $z_i, z_{i+\frac{1}{2}}, i \in \mathbb{N}$ are in a bounded convex set due to Lemma 14, we have $a = 4$ under Assumption 4 in a constrained game. The theorem is obtained by directly combining the equation above with Lemma 4. □

## B.5 PROOF OF THEOREM 2

Cases where $T < 3$ are trivial with Big $O$ notation since $a(x) = O(b(x))$ is true if $x$ belongs to a limited set, $\forall a(x) \geq 0$ and $b(x) > 0$. Suppose $T = T_1 + T_2 + 1, T_1, T_2 \in \mathbb{N}^*$ for $T \geq 2$ and $T_1 = T_2 + 2$ or $T_2 + 1$. According to Theorem 5, it holds that

$$r^{tan}(z_T) = r^{tan}(z_{T_1+T_2+1})$$

$$\leq \sqrt{\left( \frac{C_2}{\sqrt{T_2}} + C_3 \max_{k \in [T_1, T_1+T_2]} \|G_k\| \right)^2 + \frac{4D_{\mathcal{Z}} + 4\eta \max \|F_\infty\|}{\eta} \sum_{k=T_1}^{\infty} \max \|G_k\|}$$

$$= \sqrt{O\left( \frac{1}{T_2} \right) + O\left( \max_{k \in [T_1, T_1+T_2]} \|G_k\| \right)^2 + O\left( \sum_{k=T_1}^{\infty} \max \|G_k\| \right)}$$

$$= O\left( \frac{1}{\sqrt{T_2}} \right) + O\left( \max_{k \in [T_1, T_1+T_2]} \|G_k\| \right) + O\left( \sqrt{\sum_{k=T_1}^{\infty} \max \|G_k\|} \right)$$

Hence,

$$r^{tan}(z_T) = O\left( \max \left\{ \frac{1}{\sqrt{T}}, \sqrt{\sum_{k=T/2}^{\infty} \max \|G_k\|}, \max \|G_T\| \right\} \right)$$

Then, according to Lemma 3, $\frac{P_{\mathcal{G},D}(z)}{D} \leq r_{\mathcal{G}}^{tan}(z)$ and $\frac{T_{\mathcal{G},D}(z)}{\sqrt{N}D} \leq r_{\mathcal{G}}^{tan}(z)$. Hence, it holds that

$$\max \left\{ r^{tan}(z_T), \frac{T_{\mathcal{G},D}(z_T)}{\sqrt{N}D}, \frac{P_{\mathcal{G},D}(z_T)}{D} \right\}$$

$$= O\left( \max \left\{ \frac{1}{\sqrt{T}}, \sqrt{\sum_{k=T/2}^{\infty} \max \|G_k\|}, \max \|G_T\| \right\} \right)$$

Besides, we have

$$r^{tan}\left(z_T\right) = r^{tan}\left(z_{T_1+T_2}\right)$$

$$\leq \sqrt{\left(\frac{C_2}{\sqrt{T_2}} + \max_{k\in[T_1,T_1+T_2]} L_{G_k} D_0\right)^2 + \frac{4D + a\eta\max\|F_\infty\|}{\eta}\sum_{k=T_1}^\infty \max\|G_k\|}$$

$$= \sqrt{O\left(\frac{1}{T_2}\right) + O\left(\max_{k\in[T_1,T_1+T_2]} L_{G_k} D_0\right)^2 + O\left(\sum_{k=T_1}^\infty \max\|G_k\|\right)}$$

$$= O\left(\frac{1}{\sqrt{T_2}}\right) + O\left(\max_{k\in[T_1,T_1+T_2]} L_{G_k} D_0\right) + O\left(\sqrt{\sum_{k=T_1}^\infty \max\|G_k\|}\right)$$

under Assumption 4. Hence,

$$r^{tan}\left(z_T\right) = O\left(\max\left\{\frac{1}{\sqrt{T}}, \max_{k\geq T/2} L_{G_k}, \sqrt{\sum_{k=T/2}^\infty \max\|G_k\|}\right\}\right)$$

Then, according to Lemma 3, $\frac{P_{\mathcal{G},D}(z)}{D} \leq r_{\mathcal{G}}^{tan}(z)$ and $\frac{T_{\mathcal{G},D}(z)}{\sqrt{N}D} \leq r_{\mathcal{G}}^{tan}(z)$. Hence, it holds that

$$\max\left\{r^{tan}\left(z_T\right), \frac{T_{\mathcal{G},D}(z_T)}{\sqrt{N}D}, \frac{P_{\mathcal{G},D}(z_T)}{D}\right\}$$

$$= O\left(\max\left\{\frac{1}{\sqrt{T}}, \max_{k\geq T/2} L_{G_k}, \sqrt{\sum_{k=T/2}^\infty \max\|G_k\|}\right\}\right)$$

$\square$

## C  OMITTED PROOFS FOR LAST-ITERATE CONVERGENCE OF THE OG ALGORITHM

In this section, we provide detailed proof for the last-iterate convergence rate of the OG algorithm. $\forall k \geq 0$,

$$w_{k+1} = \Pi_{\mathcal{Z}}\left[z_k - \eta F_k\left(w_k\right)\right], z_{k+1} = \Pi_{\mathcal{Z}}\left[z_k - \eta F_k\left(w_{k+1}\right)\right] \tag{15}$$

The proof of last-iterate convergence of the OG algorithm is similar to that of the EG algorithm. This appendix consists of the following parts. This section consists of the following parts. First, with a method inspired by Hsieh et al. (2019); Wei et al. (2021), the best-iterate convergence rate of $\Delta(z_k, w_k)$ for the OG algorithm is proved. Then, the upper bound of tangent residual of $z_k$ is proved to be $\max\{O(\|z_k - z_{k+\frac{1}{2}}\|), O(\max\|G_k\|)\}$ if $G_k$ is $L_{G_k}$-Lipschitz and to be $O(\|z_k - z_{k+\frac{1}{2}}\|)$ no matter whether $G_k$ is $L_{G_k}$-Lipschitz. Next, the tangent residual is proved to be either non-increasing or increasing slowly enough across iterates of the OG algorithm. Finally, the last-iterate convergence rate of the OG algorithm is concluded from the conditions above.

**Lemma 16** *For any closed convex set $\mathcal{Z} \subseteq \mathbb{R}^n$, if $z_1 = \Pi_{\mathcal{Z}}[z_2 - \eta F_{k_{z_3}}(z_3)]$, then we have*

$$\|z_1 - z_2\|^2 \leq 2\eta^2 \Delta(z_2, z_3) + 4\eta^2 L \max\|z_2 - z_3\| \left\|G_{k_{z_3}}\right\| + 2\eta^2 \left\|G_{k_{z_3}}\right\|^2$$

*Proof.* For $z_4 = \Pi_{\mathcal{Z}}[z_2 - \eta F_\infty(z_2)]$, there exists

$$\|z_1 - z_2\| \leq \|z_1 - z_4\| + \|z_4 - z_2\|$$

Since $z_1 = \Pi_{\mathcal{Z}}[z_2 - \eta F_{k_{z_3}}(z_3)]$ exists and the projection operator is non-expansive, there exists

$$\|z_1 - z_4\| \leq \eta\|F_\infty(z_2) - F_\infty(z_3)\| + \eta\left\|G_{k_{z_3}}(z_3)\right\|$$

Due to Definition 7 and Lemma 11, there exists

$$\|z_2 - z_4\| = r_{\eta F, \mathcal{Z}}^{nat}(z_2) \leq r_{\eta F, \mathcal{Z}}^{tan}(z_2) = \eta r^{tan}(z_2)$$

Due to $(a+b)^2 \leq 2a^2 + 2b^2$, there exists

$$\|z_1 - z_2\|^2 \leq 2\eta^2 r^{tan}(z_2)^2 + 2\eta^2\|F_\infty(z_2) - F_\infty(z_3)\|^2 + 4\eta^2 L \max\|z_2 - z_3\| \left\|G_{k_{z_3}}\right\| + 2\eta^2 \left\|G_{k_{z_3}}\right\|^2$$

$$= 2\eta^2 \Delta(z_2, z_3) + 4\eta^2 L \max\|z_2 - z_3\| \left\|G_{k_{z_3}}\right\| + 2\eta^2 \left\|G_{k_{z_3}}\right\|^2$$

$\square$

## C.1 PREPARATION FOR ANALYZING THE BEST-ITERATE CONVERGENCE BEHAVIOURS OF THE OG ALGORITHM

Inspired by the proof of the best-iterate convergence rate with the OG algorithm in time-invariant games Hsieh et al. (2019); Wei et al. (2021), we provide Lemma 18 to analyze the best-iterate convergence behaviours of the OG algorithm with Lemma 17.

**Lemma 17** *For a game $\mathcal{G}$ converging to a monotone game with any closed convex set $\mathcal{Z} \subseteq \mathbb{R}^n$ and monotone and $L$-Lipschitz operator $F_\infty : \mathcal{Z} \to \mathbb{R}^n$ and $z_k, w_k \in \mathcal{Z}$, there exists*

$$\|w_k - w_{k+1}\|^2 \leq 2(4\eta^2 L^2)^k \|w_0 - z_0\|^2 + 2\sum_{i=0}^{k}(4\eta^2 L^2)^i \|z_{k-i} - w_{k+1-i}\|^2$$
$$+ 16\eta^2 \sum_{i=0}^{k-1}(4\eta^2 L^2)^i \|G_{k-1-i}\|^2$$

*and when $T \geq 1$, if $\eta \in \left(0, \frac{1}{2L}\right)$,*

$$\sum_{k=0}^{T} \|w_k - w_{k+1}\|^2 \leq \frac{2}{1 - 4\eta^2 L^2}\left(\|w_0 - z_0\|^2 + \sum_{k=0}^{T} \|z_k - w_{k+1}\|^2\right.$$
$$\left. + 8\eta^2 \sum_{i=0}^{T-1} \max \|G_i\|^2\right)$$

*while under Assumption 4, there exists*

$$\|w_k - w_{k+1}\|^2 \leq 2\|z_k - w_{k+1}\|^2 + \sum_{i=1}^{k} 2\prod_{j=1}^{i}\left(2\eta^2\left(L + L_{G_{k-j}}\right)^2\right)\|z_{k-i} - w_{k-i+1}\|^2$$
$$+ 2\prod_{i=1}^{k}\left(2\eta^2\left(L + L_{G_{i-1}}\right)^2\right)\|w_0 - z_0\|^2$$

$$\sum_{k=0}^{T} \|w_k - w_{k+1}\|^2 \leq E_G\left(\|w_0 - z_0\|^2 + \sum_{k=0}^{T} \|z_k - w_{k+1}\|^2\right)$$

*where*

$$E_G = 2 + \lim_{T\to\infty} 2\sum_{k=1}^{T}\left(2\eta^2\left(L + L_G\right)^2\right)^k$$

*Proof.* $\forall k \geq 1$,

$$\|w_k - z_k\|^2 \leq \eta^2 \|F_{k-1}(w_{k-1}) - F_{k-1}(w_k)\|^2$$
$$\leq \eta^2\left(L\|w_{k-1} - w_k\| + \|G_{k-1}(w_{k-1}) - G_{k-1}(w_k)\|\right)^2$$
$$\leq 2\eta^2\left(L^2\|w_{k-1} - w_k\|^2 + 4\|G_{k-1}\|^2\right)$$

$\forall k \geq 0$ there exists

$$\|w_k - w_{k+1}\|^2 \leq 2\|w_k - z_k\|^2 + 2\|z_k - w_{k+1}\|^2$$

Hence, $\forall k \geq 2$ and under Assumption 2, there exists

$$\|w_k - w_{k+1}\|^2 \leq 2\eta^2 L^2 \|w_{k-1} - w_k\|^2 + 2\|z_k - w_{k+1}\|^2 + 2\eta^2 \|G_{k-1}(w_{k-1}) - G_{k-1}(w_k)\|^2$$
$$+ 4\eta^2 L\|w_{k-1} - w_k\|\|G_{k-1}(w_{k-1}) - G_{k-1}(w_k)\|$$
$$\leq 4\eta^2 L^2\|w_{k-1} - w_k\|^2 + 2\|z_k - w_{k+1}\|^2 + 4\eta^2\|G_{k-1}(w_{k-1}) - G_{k-1}(w_k)\|^2$$
$$\leq 4\eta^2 L^2\|w_{k-1} - w_k\|^2 + 2\|z_k - w_{k+1}\|^2 + 16\eta^2\|G_{k-1}\|^2$$
$$\leq 4\eta^2 L^2\left(2(4\eta^2 L^2)^{k-1}\|w_0 - z_0\|^2 + 2\sum_{i=0}^{k-1}(4\eta^2 L^2)^i\|z_{k-1-i} - w_{k-i}\|^2\right.$$
$$\left. + 16\eta^2\sum_{i=0}^{k-2}(4\eta^2 L^2)^i\|G_{k-2-i}\|^2\right) + 2\|z_k - w_{k+1}\|^2 + 16\eta^2\|G_{k-1}\|^2$$
$$= 2(4\eta^2 L^2)^k\|w_0 - z_0\|^2 + 2\sum_{i=0}^{k}(4\eta^2 L^2)^i\|z_{k-i} - w_{k+1-i}\|^2$$
$$+ 16\eta^2\sum_{i=0}^{k-1}(4\eta^2 L^2)^i\|G_{k-1-i}\|^2$$

and when $k = 1$,

$$\|w_k - w_{k+1}\|^2 \le 2(4\eta^2 L^2)^k \|w_0 - z_0\|^2 + 2\sum_{i=0}^{k}(4\eta^2 L^2)^i \|z_{k-i} - w_{k+1-i}\|^2$$
$$+ 16\eta^2 \sum_{i=0}^{k-1}(4\eta^2 L^2)^i \|G_{k-1-i}\|^2$$

Hence, there exists

$$\sum_{k=0}^{T} \|w_k - w_{k+1}\|^2 \le \frac{2}{1 - 4\eta^2 L^2}\left(\|w_0 - z_0\|^2 + \sum_{k=0}^{T} \|z_k - w_{k+1}\|^2\right.$$
$$\left. + 8\eta^2 \sum_{i=0}^{T-1} \max\|G_i\|^2\right)$$

If $G_k$ is $L_{G_k}$-Lipschitz, $\forall k \ge 1$, there exists

$$\|w_k - z_k\|^2 \le \eta^2 \left(L + L_{G_{k-1}}\right)^2 \|w_{k-1} - w_k\|^2$$

Hence, under Assumption 4,

$$\|w_k - w_{k+1}\|^2 \le 2\eta^2 \left(L + L_{G_{k-1}}\right)^2 \|w_{k-1} - w_k\|^2 + 2\|z_k - w_{k+1}\|^2$$
$$\le 2\|z_k - w_{k+1}\|^2 + \sum_{i=0}^{k-2} 2\prod_{j=0}^{i}\left(2\eta^2 \left(L + L_{G_{k-1-j}}\right)^2\right)\|z_{k-1-i} - w_{k-i}\|^2$$
$$+ \prod_{i=1}^{k}\left(2\eta^2 \left(L + L_{G_{i-1}}\right)^2\right)\|w_0 - w_1\|^2$$
$$\le 2\|z_k - w_{k+1}\|^2 + \sum_{i=1}^{k} 2\prod_{j=1}^{i}\left(2\eta^2 \left(L + L_{G_{k-j}}\right)^2\right)\|z_{k-i} - w_{k-i+1}\|^2$$
$$+ 2\prod_{i=1}^{k}\left(2\eta^2 \left(L + L_{G_{i-1}}\right)^2\right)\|w_0 - z_0\|^2$$

By the equation above, there exists

$$\sum_{k=0}^{T} \|w_k - w_{k+1}\|^2 \le 2\left(1 + \sum_{k=1}^{T}\prod_{i=1}^{k}\left(2\eta^2 \left(L + L_{G_{i-1}}\right)^2\right)\right)\|w_0 - z_0\|^2$$
$$+ 2\|z_0 - w_1\|^2 + \sum_{k=1}^{T}\left(2\|z_k - w_{k+1}\|^2\right.$$
$$\left. + \sum_{i=1}^{k} 2\prod_{j=1}^{i}\left(2\eta^2 \left(L + L_{G_{k-j}}\right)^2\right)\|z_{k-i} - w_{k-i+1}\|^2\right)$$
$$= 2\left(1 + \sum_{k=1}^{T}\prod_{i=1}^{k}\left(2\eta^2 \left(L + L_{G_{i-1}}\right)^2\right)\right)\|w_0 - z_0\|^2$$
$$+ \sum_{k=0}^{T} \|z_k - w_{k+1}\|^2 \left(2 + \sum_{i=1}^{T-k} 2\prod_{j=1}^{i}\left(2\eta^2 \left(L + L_{G_{k+i-j}}\right)^2\right)\right)$$

Since $E_G = 2 + \lim_{T \to \infty} 2\sum_{k=1}^{T}(2\eta^2(L + L_G)^2)^k$ so that $2 + 2\sum_{k=1}^{T}\prod_{i=1}^{k}(2\eta^2(L + L_{G_{i-1}})^2) \le E_G$,

$$\sum_{k=0}^{T} \|w_k - w_{k+1}\|^2 \le E_G\left(\|w_0 - z_0\|^2 + \sum_{k=0}^{T} \|z_k - w_{k+1}\|^2\right)$$

$\square$

**Lemma 18** *For a game $\mathcal{G}$ converging to a monotone game with any closed convex set $\mathcal{Z} \subseteq \mathbb{R}^n$ and monotone and $L$-Lipschitz operator $F_\infty : \mathcal{Z} \to \mathbb{R}^n$ and $z_k, w_k \in \mathcal{Z}$, if $\eta \in \left(0, \frac{1}{\sqrt{6}L}\right)$, there exists*

$$\|z_{T+1} - z^*\|^2 \le \|z_0 - z^*\|^2 + \frac{2\eta^2 L^2}{1 - 4\eta^2 L^2}\|w_0 - z_0\|^2 - \frac{1 - 6\eta^2 L^2}{1 - 4\eta^2 L^2}\sum_{k=0}^{T} \|z_k - w_{k+1}\|^2$$
$$+ \frac{16\eta^4 L^2}{1 - 4\eta^2 L^2}\sum_{k=0}^{T-1} \max\|G_k\|^2 + 4\eta D_{\mathcal{Z}}\sum_{k=0}^{T} \max\|G_k\|$$

*and if $G_k$ is $L_{G_k}$-Lipschitz and $\eta \in \left(0, \frac{1}{\sqrt{6L^2 + 4LL_G + 2L_G^2}}\right)$, there exists*

$$\|z_{T+1} - z^*\|^2 \leq \|z_0 - z^*\|^2 + \frac{2\eta^2 (L + L_G)^2}{1 - 4\eta^2 L^2} \|w_0 - z_0\|^2 - \frac{1 - 6\eta^2 L^2 - 4\eta^2 LL_G - 2\eta^2 L_G^2}{1 - 4\eta^2 L^2}$$

$$\cdot \sum_{k=0}^{T} \|z_k - w_{k+1}\|^2 + \frac{16\eta^4 (L + L_G)^2}{1 - 4\eta^2 L^2} \sum_{k=0}^{T-1} \max \|G_k\|^2$$

$$+ 2\eta D_{\mathcal{Z}} \sum_{k=0}^{T} \max \|G_k\|$$

*while under Assumption 4, if $\eta \in \left(0, \min\left\{\frac{1}{2(L + L_G)}, \frac{1}{4L_G}\right\}\right)$,*

$$\|z_{T+1} - z^*\|^2 \leq E_{G2} \|z_0 - z^*\|^2 + E_{G2}\eta^2 (L + L_G)^2 E_G \|w_0 - z_0\|^2$$

$$- (1 - 4\eta L_G - E_G E_{G2}\eta^2 (L + L_G)^2) \sum_{k=0}^{T} \|z_k - w_{k+1}\|^2$$

*where*

$$E_G = 2 + \lim_{T \to \infty} 2 \sum_{k=1}^{T} \left(2\eta^2 (L + L_G)^2\right)^k$$

$$E_{G2} = \lim_{T \to \infty} \prod_{k=0}^{T} (1 + 4\eta L_{G_k})$$

*Proof.* $\forall k \geq 0$, there exists

$$\|z_{k+1} - z^*\|^2 = \|z_k - z^*\|^2 + \|z_{k+1} - z_k\|^2 + 2 \langle z_{k+1} - z_k, z_k - z^* \rangle$$
$$= \|z_k - z^*\|^2 - \|z_{k+1} - z_k\|^2 + 2 \langle z_{k+1} - z_k, z_{k+1} - z^* \rangle$$
$$\leq \|z_k - z^*\|^2 - \|z_{k+1} - z_k\|^2 - 2\eta \langle F_\infty(w_{k+1}), z_{k+1} - z^* \rangle$$
$$- 2\eta \langle G_k(w_{k+1}), z_{k+1} - z^* \rangle$$

Due to Lemma 17, there exists

$$\|z_{k+1} - z^*\|^2 \leq \|z_k - z^*\|^2 - \|z_{k+1} - z_k\|^2 - 2\eta \langle F_\infty(w_{k+1}), z_{k+1} - z^* \rangle$$
$$- 2\eta \langle G_k(w_{k+1}), z_{k+1} - z^* \rangle$$
$$= \|z_k - z^*\|^2 - \|z_{k+1} - z_k\|^2 - 2\eta \langle F_\infty(w_{k+1}), z_{k+1} - w_{k+1} \rangle$$
$$+ 2\eta \langle F_\infty(w_{k+1}), z^* - w_{k+1} \rangle - 2\eta \langle G_k(w_{k+1}), z_{k+1} - w_{k+1} \rangle$$
$$+ 2\eta \langle G_k(w_{k+1}), z^* - w_{k+1} \rangle$$
$$\leq \|z_k - z^*\|^2 - \|z_{k+1} - z_k\|^2 - 2\eta \langle F_\infty(w_{k+1}), z_{k+1} - w_{k+1} \rangle$$
$$- 2\eta \langle G_k(w_{k+1}), z_{k+1} - w_{k+1} \rangle + 2\eta \langle G_k(w_{k+1}), z^* - w_{k+1} \rangle$$

$\forall k \geq 0$, there exists

$$\|z_{k+1} - w_{k+1}\|^2 = \|z_{k+1} - z_k\|^2 + \|z_k - w_{k+1}\|^2 + 2 \langle z_k - w_{k+1}, z_{k+1} - z_k \rangle$$
$$= \|z_{k+1} - z_k\|^2 - \|z_k - w_{k+1}\|^2 + 2 \langle z_k - w_{k+1}, z_{k+1} - w_{k+1} \rangle$$
$$\leq \|z_{k+1} - z_k\|^2 - \|z_k - w_{k+1}\|^2 + 2\eta \langle F_\infty(w_k), z_{k+1} - w_{k+1} \rangle$$
$$+ 2\eta \langle G_k(w_k), z_{k+1} - w_{k+1} \rangle$$

By adding the two equations above, we obtain

$$\|z_{k+1} - z^*\|^2 \leq \|z_k - z^*\|^2 - \|z_{k+1} - w_{k+1}\|^2 - \|z_k - w_{k+1}\|^2$$
$$+ 2\eta \langle F_\infty(w_k) - F_\infty(w_{k+1}), z_{k+1} - w_{k+1} \rangle + 2\eta \langle G_k(w_{k+1}), z^* - w_{k+1} \rangle$$
$$+ 2\eta \langle G_k(w_k) - G_k(w_{k+1}), z_{k+1} - w_{k+1} \rangle$$
$$\leq \|z_k - z^*\|^2 - \|z_{k+1} - w_{k+1}\|^2 - \|z_k - w_{k+1}\|^2$$
$$+ 2\eta \|F_\infty(w_k) - F_\infty(w_{k+1})\| \|z_{k+1} - w_{k+1}\| + 2\eta \langle G_k(w_{k+1}), z^* - w_{k+1} \rangle$$
$$+ 2\eta \|G_k(w_k) - G_k(w_{k+1})\| \|z_{k+1} - w_{k+1}\|$$

If $\mathcal{Z}$ is bounded,

$$\begin{aligned}
\|z_{k+1} - z^*\|^2 &\leq \|z_k - z^*\|^2 - \|z_k - w_{k+1}\|^2 + \eta^2 L^2 \|w_k - w_{k+1}\|^2 + 2\eta \max\|G_k\|\|z_{k+1} - z^*\| \\
&\quad + 2\eta D_{\mathcal{Z}}\|G_k\| \\
&\leq \|z_k - z^*\|^2 - \|z_k - w_{k+1}\|^2 + \eta^2 L^2 \|w_k - w_{k+1}\|^2 + 4\eta D_{\mathcal{Z}} \max\|G_k\|
\end{aligned}$$

Then, there exists

$$\begin{aligned}
\|z_{T+1} - z^*\|^2 &\leq \|z_0 - z^*\|^2 - \sum_{k=0}^{T} \|z_k - w_{k+1}\|^2 + \eta^2 L^2 \sum_{k=0}^{T} \|w_k - w_{k+1}\|^2 \\
&\quad + 4\eta D_{\mathcal{Z}} \sum_{k=0}^{T} \max\|G_k\| \\
&\leq \|z_0 - z^*\|^2 - \sum_{k=0}^{T} \|z_k - w_{k+1}\|^2 + 4\eta D_{\mathcal{Z}} \sum_{k=0}^{T} \max\|G_k\| \\
&\quad + \frac{2\eta^2 L^2}{1 - 4\eta^2 L^2}\left(\|w_0 - z_0\|^2 + \sum_{k=0}^{T} \|z_k - w_{k+1}\|^2 + 8\eta^2 \sum_{k=0}^{T-1} \max\|G_k\|^2\right) \\
&\leq \|z_0 - z^*\|^2 + \frac{2\eta^2 L^2}{1 - 4\eta^2 L^2}\|w_0 - z_0\|^2 - \frac{1 - 6\eta^2 L^2}{1 - 4\eta^2 L^2}\sum_{k=0}^{T}\|z_k - w_{k+1}\|^2 \\
&\quad + \frac{16\eta^4 L^2}{1 - 4\eta^2 L^2}\sum_{k=0}^{T-1}\max\|G_k\|^2 + 4\eta D_{\mathcal{Z}}\sum_{k=0}^{T}\max\|G_k\|
\end{aligned}$$

If $G_k$ is $L_{G_k}$-Lipschitz, there exists

$$\begin{aligned}
\|z_{k+1} - z^*\|^2 &\leq \|z_k - z^*\|^2 - \|z_{k+1} - w_{k+1}\|^2 - \|z_k - w_{k+1}\|^2 + 2\eta L \|w_k - w_{k+1}\| \|z_{k+1} - w_{k+1}\| \\
&\quad + 2\eta \langle G_k(w_{k+1}), z^* - w_{k+1}\rangle + 2\eta L_{G_k} \|w_k - w_{k+1}\| \|z_{k+1} - w_{k+1}\| \\
&\leq \|z_k - z^*\|^2 - \|z_k - w_{k+1}\|^2 + \eta^2 (L + L_{G_k})^2 \|w_k - w_{k+1}\|^2 + 2\eta \langle G_k(w_{k+1}), z^* - w_{k+1}\rangle \\
&\leq \|z_k - z^*\|^2 - \|z_k - w_{k+1}\|^2 + \eta^2 (L + L_{G_k})^2 \|w_k - w_{k+1}\|^2 + 2\eta D_{\mathcal{Z}} \max\|G_k(w_{k+1})\|
\end{aligned}$$

Hence, there exists

$$\begin{aligned}
\|z_{T+1} - z^*\|^2 &\leq \|z_0 - z^*\|^2 - \sum_{k=0}^{T} \|z_k - w_{k+1}\|^2 + \eta^2 (L + L_G)^2 \sum_{k=0}^{T} \|w_k - w_{k+1}\|^2 \\
&\quad + 2\eta D_{\mathcal{Z}} \sum_{k=0}^{T} \max\|G_k\| \\
&\leq \|z_0 - z^*\|^2 - \sum_{k=0}^{T} \|z_k - w_{k+1}\|^2 + \frac{2\eta^2 (L + L_G)^2}{1 - 4\eta^2 L^2}\left(\|w_0 - z_0\|^2 + \sum_{k=0}^{T} \|z_k - w_{k+1}\|^2\right. \\
&\quad \left. + 8\eta^2 \sum_{i=0}^{T-1} \max\|G_i\|^2\right) + 2\eta D_{\mathcal{Z}} \sum_{k=0}^{T} \max\|G_k\| \\
&\leq \|z_0 - z^*\|^2 + \frac{2\eta^2 (L + L_G)^2}{1 - 4\eta^2 L^2}\|w_0 - z_0\|^2 - \frac{1 - 6\eta^2 L^2 - 4\eta^2 L L_G - 2\eta^2 L_G^2}{1 - 4\eta^2 L^2} \\
&\quad \cdot \sum_{k=0}^{T}\|z_k - w_{k+1}\|^2 + \frac{16\eta^4 (L + L_G)^2}{1 - 4\eta^2 L^2}\sum_{k=0}^{T-1}\max\|G_k\|^2 + 2\eta D_{\mathcal{Z}}\sum_{k=0}^{T}\max\|G_k\|
\end{aligned}$$

Under Assumption 4, there exists

$$\begin{aligned}
\|z_{k+1} - z^*\|^2 &\leq \|z_k - z^*\|^2 - \|z_k - w_{k+1}\|^2 + \eta^2 (L + L_{G_k})^2 \|w_k - w_{k+1}\|^2 + 2\eta L_{G_k} \|z^* - w_{k+1}\|^2 \\
&\leq (1 + 4\eta L_{G_k}) \|z_k - z^*\|^2 - (1 - 4\eta L_{G_k}) \|z_k - w_{k+1}\|^2 + \eta^2 (L + L_{G_k})^2 \|w_k - w_{k+1}\|^2
\end{aligned}$$

Hence,

$$
\begin{aligned}
\|z_{T+1} - z^*\|^2 &\leq \prod_{k=0}^{T} (1 + 4\eta L_{G_k}) \|z_0 - z^*\|^2 - \sum_{k=0}^{T-1} \prod_{i=k+1}^{T} (1 + 4\eta L_{G_i})(1 - 4\eta L_{G_k}) \|z_k - w_{k+1}\|^2 \\
&\quad - (1 - 4\eta L_{G_T}) \|z_T - w_{T+1}\|^2 + \sum_{k=0}^{T-1} \prod_{i=k+1}^{T} (1 + 4\eta L_{G_i}) \eta^2 (L + L_{G_k})^2 \|w_k - w_{k+1}\|^2 \\
&\quad + \eta^2 (L + L_{G_T})^2 \|w_T - w_{T+1}\|^2 \\
&\leq E_{G2} \|z_0 - z^*\|^2 - \sum_{k=0}^{T} (1 - 4\eta L_{G_k}) \|z_k - w_{k+1}\|^2 \\
&\quad + E_{G2} \eta^2 (L + L_G)^2 \sum_{k=0}^{T} \|w_k - w_{k+1}\|^2 \\
&\leq E_{G2} \|z_0 - z^*\|^2 - \sum_{k=0}^{T} (1 - 4\eta L_{G_k}) \|z_k - w_{k+1}\|^2 \\
&\quad + E_{G2} \eta^2 (L + L_G)^2 E_G \left( \|w_0 - z_0\|^2 + \sum_{k=0}^{T} \|z_k - w_{k+1}\|^2 \right) \\
&\leq E_{G2} \|z_0 - z^*\|^2 + E_{G2} \eta^2 (L + L_G)^2 E_G \|w_0 - z_0\|^2 \\
&\quad - (1 - 4\eta L_G - E_G E_{G2} \eta^2 (L + L_G)^2) \sum_{k=0}^{T} \|z_k - w_{k+1}\|^2
\end{aligned}
$$

$\square$

## C.2 Best-iterate convergence of $\Delta(z_k, w_k)$

This section shows that $\forall T \in \mathbb{N}^*$, there exists $t^* \in [[T]]$ satisfying $\Delta(z_{t^*}, w_{t^*}) = O(1/T)$.

Following is the proof of the boundedness of any sequence of players' actions with the OG algorithm in $(L + L_{G_k})$-Lipschitz games described under Assumption 4.

**Lemma 19** *For a game $\mathcal{G}$ converging to a monotone game with any closed convex set $\mathcal{Z} \in \mathbb{R}^n$, monotone $L$-Lipschitz operator $F_\infty$ mapping from $\mathcal{Z}$ to $\mathbb{R}^n$ and $z_k \in \mathcal{Z}$, with the OG algorithm under Assumption 4, there exists $C_z, C_w > 0$ so that $\|z_k - z^*\| \leq C_z$ and $\|w_k - z^*\| \leq C_w$.*

*Proof.* Under Assumption 4, with Lemma 18, we find that $\forall t \in \mathbb{N}^*$, $\|z_k - z^*\|^2$ is bounded in any sequence of games in this paper with the OG algorithm. Hence, there exists $C_z > 0$ so that $\|z_k - z^*\| < C_z$.

With OG algorithm, we have $z_k = \Pi_{\mathcal{Z}}[z_{k-1} - \eta F_k(w_k)]$. Hence, for $C_w = 2C_z$,

$$
\begin{aligned}
\|w_{k+1} - z_k\| &= \|\Pi_{\mathcal{Z}}[z_k - \eta F_k(w_k)] - \Pi_{\mathcal{Z}}[z_{k-1} - \eta F_k(w_k)]\| \\
&\leq \|z_k - z_{k-1}\| \leq \|z_k - z^*\| + \|z^* - z_{k-1}\| \leq 2C_z \leq C_w
\end{aligned}
$$

$\square$

**Lemma 20** *For a game $\mathcal{G}$ converging to a monotone game with any closed convex set $\mathcal{Z} \subseteq \mathbb{R}^n$ and monotone and $L$-Lipschitz operator $F_\infty : \mathcal{Z} \to \mathbb{R}^n$ and $z_k, w_k \in \mathcal{Z}$, if $\eta \in \left(0, \frac{1}{\sqrt{6}L}\right)$, there exists*

$$
\begin{aligned}
\sum_{k=1}^{T} \eta^2 \Delta(z_k, w_k) &\leq \frac{4 + 20\eta^4 L^4}{1 - 6\eta^2 L^2} \|z_0 - z^*\|^2 + \frac{4\eta^2 L^2 (5\eta^2 L^2 + 6)}{1 - 6\eta^2 L^2} \|w_0 - z_0\|^2 \\
&\quad + \frac{4\eta^2 (9 + 4\eta^2 L^2 - 20\eta^4 L^4)}{1 - 6\eta^2 L^2} E_{mk^2} + \frac{4\eta D_{\mathcal{Z}} (4 + 20\eta^4 L^4)}{1 - 6\eta^2 L^2} E_{mk}
\end{aligned}
$$

*where*

$$
\begin{aligned}
E_{mk} &= \sum_{k=0}^{\infty} \max \|G_k\| \\
E_{mk^2} &= \sum_{k=0}^{\infty} \max \|G_k\|^2
\end{aligned}
$$

*Under Assumption 3, if $\eta \in \left(0, \frac{1}{\sqrt{6L^2+4LL_G+2L_G^2}}\right)$, there exists*

$$\sum_{k=1}^{T} \eta^2 \Delta(z_k, w_k) \leq \frac{4(1-4\eta^2 L^2) + 2\eta^2 L^2 \left(\eta^2 (L+L_G)^2 + 8 + 8\eta^2 L^2\right)}{1 - 6\eta^2 L^2 - 4\eta^2 LL_G - 2\eta^2 L_G^2} \|z_0 - z^*\|^2 + \|w_0 - z_0\|^2$$

$$\cdot \left(\frac{2\eta^2 (L+L_G)^2 \left(4 + 2\eta^2 L^2 \left(8\eta^2 L^2 + \eta^2 (L+L_G)^2\right)\right)}{(1 - 6\eta^2 L^2 - 4\eta^2 LL_G - 2\eta^2 L_G^2)(1 - 4\eta^2 L^2)} + \frac{2\eta^2 L^2 \left(8 + 8\eta^2 L^2 + \eta^2 (L+L_G)^2\right)}{1 - 4\eta^2 L^2}\right)$$

$$+ \frac{8\eta D_{\mathcal{Z}}(1-4\eta^2 L^2) + 4\eta^3 D_{\mathcal{Z}} L^2 \left(\eta^2 (L+L_G)^2 + 8 + 8\eta^2 L^2\right)}{1 - 6\eta^2 L^2 - 4\eta^2 LL_G - 2\eta^2 L_G^2} E_{mk} + 4\eta^2 E_{mk^2} \left(9 + 8\eta^2 L^2\right.$$

$$+ \frac{2\eta^2 L^2 \left(8 + 8\eta^2 L^2 + \eta^2 (L+L_G)^2\right)}{1 - 4\eta^2 L^2} + \frac{2\eta^2 (L+L_G)^2 \left(4 + 2\eta^2 L^2 \left(8\eta^2 L^2 + \eta^2 (L+L_G)^2\right)\right)}{(1 - 6\eta^2 L^2 - 4\eta^2 LL_G - 2\eta^2 L_G^2)(1 - 4\eta^2 L^2)}\right)$$

*Under Assumption 4, if $1 - 4\eta L_G - E_G E_{G2}\eta^2 (L+L_G)^2 > 0$, there exists*

$$\sum_{k=1}^{T} \Delta(z_k, w_k) \leq \frac{4 + E_G \eta^2 (L+L_G)^2 \left(4 + 3\eta^2 L^2\right)}{1 - 4\eta L_G - E_G E_{G2}\eta^2 (L+L_G)^2} E_{G2} \|z_0 - z^*\|^2$$

$$+ \left(\frac{4E_{G2} + E_G E_{G2}\eta^2 (L+L_G)^2 \left(4 + 3\eta^2 L^2\right)}{1 - 4\eta L_G - E_G E_{G2}\eta^2 (L+L_G)^2} + 4 + 3\eta^2 L^2\right)(\eta^2 (L+L_G)^2 E_G) \|w_0 - z_0\|^2$$

$$+ 2\eta^2 D^2 \sum_{k=1}^{T} L_{G_{k-1}} \left(2L + L_{G_{k-1}}\right)$$

*where $D = \max_{t \in \mathbb{N}}\{\|z_t - z^*\|, \|z_t - w_t\|\}$.*

*Proof.* $\forall k \geq 1$, due to Lemma 10 , there exists

$$\eta^2 r^{tan}(z_k)^2 \leq \left(\|z_{k-1} - z_k + \eta F_\infty(z_k) - \eta F_\infty(w_k)\| + \eta \max \|G_{k-1}\|\right)^2$$

$$\leq 2\|z_{k-1} - z_k\|^2 + 2\eta^2 \left(\|F_\infty(z_k) - F_\infty(w_k)\| + \max \|G_{k-1}\|\right)^2$$

$$\leq 2\|z_{k-1} - z_k\|^2 + 2\eta^2 \left(L\|z_k - w_k\| + \max \|G_{k-1}\|\right)^2$$

$$\leq 4\|z_{k-1} - w_k\|^2 + 4\|w_k - z_k\|^2 + 4\eta^2 L^2 \|z_k - w_k\|^2 + 4\eta^2 \max \|G_{k-1}\|^2$$

$$\leq 4\|z_{k-1} - w_k\|^2 + \left(4 + 4\eta^2 L^2\right) \left(2\eta^2 L^2 \|w_{k-1} - w_k\|^2 + 8\eta^2 \max \|G_{k-1}\|^2\right)$$

$$+ 4\eta^2 \max \|G_{k-1}\|^2$$

$$= 4\|z_{k-1} - w_k\|^2 + 2\eta^2 L^2 \left(4 + 4\eta^2 L^2\right) \|w_{k-1} - w_k\|^2 + 4\eta^2 \left(9 + 8\eta^2 L^2\right) \max \|G_{k-1}\|^2$$

Hence, according to Lemma 18, when $\eta \in \left(0, \frac{1}{\sqrt{6}L}\right)$, there exists

$$\sum_{k=1}^{T} \eta^2 \Delta(z_k, w_k) = \sum_{k=1}^{T} \left(\eta^2 r^{tan}(z_k)^2 + \|\eta F_\infty(z_k) - \eta F_\infty(w_k)\|^2\right)$$

$$\leq 4 \sum_{k=1}^{T} \|z_{k-1} - w_k\|^2 + \left(4 + 5\eta^2 L^2\right) 2\eta^2 L^2 \sum_{k=1}^{T} \|w_{k-1} - w_k\|^2 + 4\eta^2 \left(9 + 10\eta^2 L^2\right) \sum_{k=1}^{T} \max \|G_{k-1}\|^2$$

$$= 4 \sum_{k=0}^{T-1} \|z_k - w_{k+1}\|^2 + \left(4 + 5\eta^2 L^2\right) 2\eta^2 L^2 \sum_{k=0}^{T-1} \|w_k - w_{k+1}\|^2 + 4\eta^2 \left(9 + 10\eta^2 L^2\right) \sum_{k=0}^{T-1} \max \|G_k\|^2$$

$$\leq 4 \sum_{k=0}^{T-1} \|z_k - w_{k+1}\|^2 + \left(4 + 5\eta^2 L^2\right) 2\eta^2 L^2 \frac{2}{1 - 4\eta^2 L^2} \left(\|w_0 - z_0\|^2 + \sum_{k=0}^{T-1} \|z_k - w_{k+1}\|^2\right)$$

$$+ 8\eta^2 \sum_{i=0}^{T-2} \max \|G_i\|^2\right) + 4\eta^2 \left(9 + 10\eta^2 L^2\right) \sum_{k=0}^{T-1} \max \|G_k\|^2$$

$$
= \left( \frac{4 + 20\eta^4 L^4}{1 - 4\eta^2 L^2} \right) \sum_{k=0}^{T-1} \|z_k - w_{k+1}\|^2 + \frac{\left(4 + 5\eta^2 L^2\right) 4\eta^2 L^2}{1 - 4\eta^2 L^2} \left( \|w_0 - z_0\|^2 + 8\eta^2 \sum_{i=0}^{T-2} \max \|G_i\|^2 \right)
$$

$$
+ 4\eta^2 \left(9 + 10\eta^2 L^2\right) \sum_{k=0}^{T-1} \max \|G_k\|^2
$$

$$
\leq \left( \frac{4 + 20\eta^4 L^4}{1 - 6\eta^2 L^2} \right) \left( \|z_0 - z^*\|^2 + \frac{2\eta^2 L^2}{1 - 4\eta^2 L^2} \|w_0 - z_0\|^2 + \frac{16\eta^4 L^2}{1 - 4\eta^2 L^2} \sum_{i=0}^{T-2} \max \|G_i\|^2 \right.
$$

$$
\left. + 4\eta D_{\mathcal{Z}} \sum_{k=0}^{T-1} \max \|G_k\| \right) + \frac{\left(4 + 5\eta^2 L^2\right) 4\eta^2 L^2}{1 - 4\eta^2 L^2} \left( \|w_0 - z_0\|^2 + 8\eta^2 \sum_{i=0}^{T-2} \max \|G_i\|^2 \right)
$$

$$
+ 4\eta^2 \left(9 + 10\eta^2 L^2\right) \sum_{k=0}^{T-1} \max \|G_k\|^2
$$

$$
\leq \left( \frac{4 + 20\eta^4 L^4}{1 - 6\eta^2 L^2} \right) \left( \|z_0 - z^*\|^2 + \frac{2\eta^2 L^2}{1 - 4\eta^2 L^2} \|w_0 - z_0\|^2 + \frac{16\eta^4 L^2}{1 - 4\eta^2 L^2} E_{mk^2} + 4\eta D_{\mathcal{Z}} E_{mk} \right)
$$

$$
+ \frac{\left(4 + 5\eta^2 L^2\right) 4\eta^2 L^2}{1 - 4\eta^2 L^2} \left( \|w_0 - z_0\|^2 + 8\eta^2 E_{mk^2} \right) + 4\eta^2 \left(9 + 10\eta^2 L^2\right) E_{mk^2}
$$

$$
\leq \frac{4 + 20\eta^4 L^4}{1 - 6\eta^2 L^2} \|z_0 - z^*\|^2 + \frac{2\eta^2 L^2 \left(4 + 20\eta^4 L^4\right) + \left(1 - 6\eta^2 L^2\right) \left(4 + 5\eta^2 L^2\right) 4\eta^2 L^2}{\left(1 - 4\eta^2 L^2\right) \left(1 - 6\eta^2 L^2\right)} \|w_0 - z_0\|^2 + E_{mk^2}
$$

$$
\cdot \frac{16\eta^4 L^2 \left(4 + 20\eta^4 L^4\right) + 32\eta^4 L^2 \left(4 + 5\eta^2 L^2\right) \left(1 - 6\eta^2 L^2\right) + 4\eta^2 \left(9 + 10\eta^2 L^2\right) \left(1 - 4\eta^2 L^2\right) \left(1 - 6\eta^2 L^2\right)}{\left(1 - 4\eta^2 L^2\right) \left(1 - 6\eta^2 L^2\right)}
$$

$$
+ \frac{4\eta D_{\mathcal{Z}} \left(4 + 20\eta^4 L^4\right)}{1 - 6\eta^2 L^2} E_{mk}
$$

$$
= \frac{4 + 20\eta^4 L^4}{1 - 6\eta^2 L^2} \|z_0 - z^*\|^2 + \frac{4\eta^2 L^2 \left(5\eta^2 L^2 + 6\right)}{1 - 6\eta^2 L^2} \|w_0 - z_0\|^2 + \frac{4\eta^2 \left(9 + 4\eta^2 L^2 - 20\eta^4 L^4\right)}{1 - 6\eta^2 L^2} E_{mk^2}
$$

$$
+ \frac{4\eta D_{\mathcal{Z}} \left(4 + 20\eta^4 L^4\right)}{1 - 6\eta^2 L^2} E_{mk}
$$

Under Assumption 3, due to Lemma 10, there exists

$$
\eta^2 r^{tan}(z_k)^2 \leq \left( \|z_{k-1} - z_k + \eta F_\infty(z_k) - \eta F_\infty(w_k)\| + \eta \max \|G_{k-1}\| \right)^2
$$

$$
\leq 4 \|z_{k-1} - w_k\|^2 + 2\eta^2 L^2 \left(4 + 4\eta^2 L^2\right) \|w_{k-1} - w_k\|^2 + 4\eta^2 \left(9 + 8\eta^2 L^2\right) \max \|G_{k-1}\|^2
$$

By adding the two inequalities above and summing them with $k \in [[T]]$, we obtain

$$
\sum_{k=1}^{T} \eta^2 \Delta(z_k, w_k) = \sum_{k=1}^{T} \left( \eta^2 r^{tan}(z_k)^2 + \|\eta F_\infty(z_k) - \eta F_\infty(w_k)\|^2 \right)
$$

$$
\leq 4 \sum_{k=1}^{T} \|z_{k-1} - w_k\|^2 + 2\eta^2 L^2 \left(4 + 4\eta^2 L^2\right) \sum_{k=1}^{T} \|w_{k-1} - w_k\|^2
$$

$$
+ 4\eta^2 \left(9 + 8\eta^2 L^2\right) \sum_{k=1}^{T} \max \|G_{k-1}\|^2
$$

$$
+ \sum_{k=1}^{T} \eta^2 L^2 \left( \eta^2 \left(L + L_{G_{k-1}}\right)^2 \|w_{k-1} - w_k\|^2 \right)
$$

$$
\leq 4 \sum_{k=1}^{T} \|z_{k-1} - w_k\|^2 + 4\eta^2 \left(9 + 8\eta^2 L^2\right) \sum_{k=1}^{T} \max \|G_{k-1}\|^2
$$

$$
+ \eta^2 L^2 \left( 2 \left(4 + 4\eta^2 L^2\right) + \eta^2 \left(L + L_G\right)^2 \right) \sum_{k=1}^{T} \|w_{k-1} - w_k\|^2
$$

$$
= 4 \sum_{k=0}^{T-1} \|z_k - w_{k+1}\|^2 + 4\eta^2 \left(9 + 8\eta^2 L^2\right) \sum_{k=0}^{T-1} \max \|G_k\|^2
$$

$$
+ \eta^2 L^2 \left( 8 + 8\eta^2 L^2 + \eta^2 \left(L + L_G\right)^2 \right) \sum_{k=0}^{T-1} \|w_k - w_{k+1}\|^2
$$

According to Lemmas 17 and 18 , there exists

$$\sum_{t=1}^{T} \eta^2 \Delta(z_k, w_k) \le 4 \sum_{k=0}^{T-1} \|z_k - w_{k+1}\|^2 + 4\eta^2 \left(9 + 8\eta^2 L^2\right) \sum_{k=0}^{T-1} \max \|G_k\|^2 + 2\eta^2 L^2$$

$$\frac{8 + 8\eta^2 L^2 + \eta^2 \left(L + L_G\right)^2}{1 - 4\eta^2 L^2} \left( \sum_{k=0}^{T-1} \|z_k - w_{k+1}\|^2 + 4\eta^2 \sum_{i=0}^{T-2} \max \|G_i\|^2 \right.$$

$$\left. + \|w_0 - z_0\|^2 \right)$$

$$\le \left( 4 + \frac{2\eta^2 L^2 \left(8 + 8\eta^2 L^2 + \eta^2 \left(L + L_G\right)^2\right)}{1 - 4\eta^2 L^2} \right)$$

$$\left( \frac{1 - 4\eta^2 L^2}{1 - 6\eta^2 L^2 - 4\eta^2 L L_G - 2\eta^2 L_G^2} \|z_0 - z^*\|^2 \right.$$

$$+ \frac{2\eta^2 \left(L + L_G\right)^2}{1 - 6\eta^2 L^2 - 4\eta^2 L L_G - 2\eta^2 L_G^2} \|w_0 - z_0\|^2$$

$$+ \frac{8\eta^4 \left(L + L_G\right)^2}{1 - 6\eta^2 L^2 - 4\eta^2 L L_G - 2\eta^2 L_G^2} \sum_{k=0}^{T-1} \max \|G_k\|^2$$

$$\left. + \frac{2\eta D_{\mathcal{Z}} \left(1 - 4\eta^2 L^2\right)}{1 - 6\eta^2 L^2 - 4\eta^2 L L_G - 2\eta^2 L_G^2} \sum_{k=0}^{T} \max \|G_k\| \right)$$

$$+ \left( 4\eta^2 \left(9 + 8\eta^2 L^2\right) + \frac{8\eta^4 L^2 \left(8 + 8\eta^2 L^2 + \eta^2 \left(L + L_G\right)^2\right)}{1 - 4\eta^2 L^2} \right) E_{mk^2}$$

$$+ \frac{2\eta^2 L^2 \left(8 + 8\eta^2 L^2 + \eta^2 \left(L + L_G\right)^2\right)}{1 - 4\eta^2 L^2} \|w_0 - z_0\|^2$$

$$\le \frac{4 \left(1 - 4\eta^2 L^2\right) + 2\eta^2 L^2 \left(8 + 8\eta^2 L^2 + \eta^2 \left(L + L_G\right)^2\right)}{1 - 6\eta^2 L^2 - 4\eta^2 L L_G - 2\eta^2 L_G^2} \|z_0 - z^*\|^2$$

$$+ 2\eta D_{\mathcal{Z}} E_{mk} \frac{4 \left(1 - 4\eta^2 L^2\right) + 2\eta^2 L^2 \left(8 + 8\eta^2 L^2 + \eta^2 \left(L + L_G\right)^2\right)}{1 - 6\eta^2 L^2 - 4\eta^2 L L_G - 2\eta^2 L_G^2}$$

$$+ E_{mk^2} \left( 4\eta^2 \left(9 + 8\eta^2 L^2\right) + \frac{8\eta^4 L^2 \left(8 + 8\eta^2 L^2 + \eta^2 \left(L + L_G\right)^2\right)}{1 - 4\eta^2 L^2} \right.$$

$$\left. + \frac{8\eta^4 \left(L + L_G\right)^2 \left(4 + 2\eta^2 L^2 \left(8\eta^2 L^2 + \eta^2 \left(L + L_G\right)^2\right)\right)}{\left(1 - 6\eta^2 L^2 - 4\eta^2 L L_G - 2\eta^2 L_G^2\right) \left(1 - 4\eta^2 L^2\right)} \right)$$

$$+ \left( \frac{2\eta^2 \left(L + L_G\right)^2 \left(4 + 2\eta^2 L^2 \left(8\eta^2 L^2 + \eta^2 \left(L + L_G\right)^2\right)\right)}{\left(1 - 6\eta^2 L^2 - 4\eta^2 L L_G - 2\eta^2 L_G^2\right) \left(1 - 4\eta^2 L^2\right)} \right.$$

$$\left. + \frac{2\eta^2 L^2 \left(8 + 8\eta^2 L^2 + \eta^2 \left(L + L_G\right)^2\right)}{1 - 4\eta^2 L^2} \right) \|w_0 - z_0\|^2$$

Under Assumption 4, due to Lemma 10, there exists

$$\eta^2 r^{tan}(z_k)^2 \le \left( \|z_{k-1} - z_k + \eta F_\infty(z_k) - \eta F_\infty(w_k)\| + \eta L_{G_{k-1}} D \right)^2$$

$$\le 2 \|z_{k-1} - z_k\|^2 + 2\eta^2 \left( L \|z_k - w_k\| + L_{G_{k-1}} D \right)^2$$

$$\le 4 \|z_{k-1} - w_k\|^2 + \left(4 + 2\eta^2 L^2\right) \|z_k - w_k\|^2 + 2\eta^2 D^2 L_{G_{k-1}} \left(2L + L_{G_{k-1}}\right)$$

Hence, with $\|w_k - z_k\|^2 \leq \eta^2 \left(L + L_{G_{k-1}}\right)^2 \|w_{k-1} - w_k\|^2$ and according to Lemmas 17 and 18, there exists

$$\sum_{k=1}^{T} \eta^2 \Delta(z_k, w_k) \leq E_G \eta^2 \left(L + L_G\right)^2 \left(4 + 3\eta^2 L^2\right) \|w_0 - z_0\|^2$$

$$+ \left(4 + E_G \eta^2 \left(L + L_G\right)^2 \left(4 + 3\eta^2 L^2\right)\right) \sum_{k=0}^{T-1} \|z_k - w_{k+1}\|^2 + 2\eta^2 D^2 \sum_{k=1}^{T} L_{G_{k-1}} \left(2L + L_{G_{k-1}}\right)$$

$$\leq E_G \eta^2 \left(L + L_G\right)^2 \left(4 + 3\eta^2 L^2\right) \|w_0 - z_0\|^2 + 2\eta^2 D^2 \sum_{k=1}^{T} L_{G_{k-1}} \left(2L + L_{G_{k-1}}\right)$$

$$+ \frac{\left(4 + E_G \eta^2 \left(L + L_G\right)^2 \left(4 + 3\eta^2 L^2\right)\right) \left(E_{G2} \|z_0 - z^*\|^2 + E_{G2} \eta^2 \left(L + L_G\right)^2 E_G \|w_0 - z_0\|^2\right)}{1 - 4\eta L_G - E_G E_{G2} \eta^2 \left(L + L_G\right)^2}$$

$$\leq \frac{4 + E_G \eta^2 \left(L + L_G\right)^2 \left(4 + 3\eta^2 L^2\right)}{1 - 4\eta L_G - E_G E_{G2} \eta^2 \left(L + L_G\right)^2} E_{G2} \|z_0 - z^*\|^2 + \|w_0 - z_0\|^2$$

$$\cdot \left(\frac{4 + E_G \eta^2 \left(L + L_G\right)^2 \left(4 + 3\eta^2 L^2\right)}{1 - 4\eta L_G - E_G E_{G2} \eta^2 \left(L + L_G\right)^2} E_{G2} \eta^2 \left(L + L_G\right)^2 E_G + E_G \eta^2 \left(L + L_G\right)^2 \left(4 + 3\eta^2 L^2\right)\right)$$

$$+ 2\eta^2 D^2 \sum_{k=1}^{T} L_{G_{k-1}} \left(2L + L_{G_{k-1}}\right)$$

$$= \frac{4 + E_G \eta^2 \left(L + L_G\right)^2 \left(4 + 3\eta^2 L^2\right)}{1 - 4\eta L_G - E_G E_{G2} \eta^2 \left(L + L_G\right)^2} E_{G2} \|z_0 - z^*\|^2 + \left(\frac{4E_{G2} + E_G E_{G2} \eta^2 \left(L + L_G\right)^2 \left(4 + 3\eta^2 L^2\right)}{1 - 4\eta L_G - E_G E_{G2} \eta^2 \left(L + L_G\right)^2}\right.$$

$$\left. + 4 + 3\eta^2 L^2\right) \left(\eta^2 \left(L + L_G\right)^2 E_G\right) \|w_0 - z_0\|^2 + 2\eta^2 D^2 \sum_{k=1}^{T} L_{G_{k-1}} \left(2L + L_{G_{k-1}}\right)$$

$\square$

From Lemma 20, we obtain the following lemma.

**Full version of Lemma 5.** *For a game $\mathcal{G}$ converging to a monotone game with any closed convex set $\mathcal{Z} \subseteq \mathbb{R}^n$ and monotone and $L$-Lipschitz operator $F_\infty : \mathcal{Z} \to \mathbb{R}^n$, let $z^*$ be a solution of the game $\mathcal{G}$. Then $\forall T \geq 1$, if $\eta \in \left(0, \frac{1}{\sqrt{6}L}\right)$, there exists $t^* \in [[T]]$ satisfying*

$$\eta^2 \Delta(z_{t^*}, w_{t^*}) \leq \frac{1}{T} \left(\frac{4 + 20\eta^4 L^4}{1 - 6\eta^2 L^2} \|z_0 - z^*\|^2 + \frac{4\eta^2 L^2 \left(5\eta^2 L^2 + 6\right)}{1 - 6\eta^2 L^2} \|w_0 - z_0\|^2\right.$$

$$\left. + \frac{4\eta^2 \left(9 + 4\eta^2 L^2 - 20\eta^4 L^4\right)}{1 - 6\eta^2 L^2} E_{mk^2} + \frac{4\eta D_{\mathcal{Z}} \left(4 + 20\eta^4 L^4\right)}{1 - 6\eta^2 L^2} E_{mk}\right)$$

*Under Assumption 3, if $\eta \in \left(0, \frac{1}{\sqrt{6L^2 + 4LL_G + 2L_G^2}}\right)$, there exists $t^* \in [[T]]$ satisfying*

$$\eta^2 \Delta(z_{t^*}, w_{t^*}) \leq \frac{1}{T} \left(\frac{4(1 - 4\eta^2 L^2) + 2\eta^2 L^2 \left(\eta^2 \left(L + L_G\right)^2 + 8 + 8\eta^2 L^2\right)}{1 - 6\eta^2 L^2 - 4\eta^2 LL_G - 2\eta^2 L_G^2} \|z_0 - z^*\|^2 + \|w_0 - z_0\|^2\right.$$

$$\cdot \left(\frac{2\eta^2 \left(L + L_G\right)^2 \left(4 + 2\eta^2 L^2 \left(8\eta^2 L^2 + \eta^2 \left(L + L_G\right)^2\right)\right)}{\left(1 - 6\eta^2 L^2 - 4\eta^2 LL_G - 2\eta^2 L_G^2\right) \left(1 - 4\eta^2 L^2\right)} + \frac{2\eta^2 L^2 \left(8 + 8\eta^2 L^2 + \eta^2 \left(L + L_G\right)^2\right)}{1 - 4\eta^2 L^2}\right)$$

$$+ \frac{8\eta D_{\mathcal{Z}}(1 - 4\eta^2 L^2) + 4\eta^3 D_{\mathcal{Z}} L^2 \left(\eta^2 \left(L + L_G\right)^2 + 8 + 8\eta^2 L^2\right)}{1 - 6\eta^2 L^2 - 4\eta^2 LL_G - 2\eta^2 L_G^2} E_{mk} + 4\eta^2 E_{mk^2} \left(9 + 8\eta^2 L^2\right.$$

$$\left. + \frac{2\eta^2 L^2 \left(8 + 8\eta^2 L^2 + \eta^2 \left(L + L_G\right)^2\right)}{1 - 4\eta^2 L^2} + \frac{2\eta^2 \left(L + L_G\right)^2 \left(4 + 2\eta^2 L^2 \left(8\eta^2 L^2 + \eta^2 \left(L + L_G\right)^2\right)\right)}{\left(1 - 6\eta^2 L^2 - 4\eta^2 LL_G - 2\eta^2 L_G^2\right) \left(1 - 4\eta^2 L^2\right)}\right)\right)$$

*while under Assumption 4, if* $1 - 4\eta L_G - E_G E_{G2}\eta^2 (L + L_G)^2 > 0$, *there exists* $t^* \in [[T]]$ *satisfying*

$$
\eta^2 \Delta(z_{t^*}, w_{t^*}) \leq \frac{1}{T} \left( \frac{4 + E_G \eta^2 (L + L_G)^2 (4 + 3\eta^2 L^2)}{1 - 4\eta L_G - E_G E_{G2}\eta^2 (L + L_G)^2} E_{G2} \|z_0 - z^*\|^2 \right.
$$

$$
+ \left( \frac{4E_{G2} + E_G E_{G2}\eta^2 (L + L_G)^2 (4 + 3\eta^2 L^2)}{1 - 4\eta L_G - E_G E_{G2}\eta^2 (L + L_G)^2} + 4 + 3\eta^2 L^2 \right) (\eta^2 (L + L_G)^2 E_G) \|w_0 - z_0\|^2
$$

$$
\left. + 2\eta^2 D^2 \sum_{k=1}^{T} L_{G_{k-1}} (2L + L_{G_{k-1}}) \right)
$$

where $D = \max_{t \in \mathbb{N}} \{\|z_t - z^*\|, \|z_t - w_t\|\}$.

*Proof.* $\forall T \geq 1$, there exists $t^* \in [[T]]$

$$
\eta^2 \Delta(z_{t^*}, w_{t^*}) \leq \frac{1}{T} \sum_{k=1}^{T} \eta^2 \Delta(z_k, w_k)
$$

The lemma is directly obtained with the inequality above from Lemma 20. $\qquad\square$

### C.3 Approximate monotonicity of $\Delta(z_k, w_k)$

This section shows that $\Delta(z_k, w_k)$ is either non-increasing or increasing at a low rate across iterates with the OG algorithm.

**Restatement of Theorem 3.** *For a game $\mathcal{G}$ converging to a monotone game, $\forall \eta \in \left(0, \frac{1}{2L}\right)$, $z_k \in \mathcal{Z}$, $\Delta(z_k, w_k) \geq \Delta(z_{k+1}, w_{k+1}) - \frac{1}{\eta}(3D_{\mathcal{Z}} + 4\eta \max \|F_\infty\|) \max \|G_k\|$, while $\Delta(z_k, w_k) \geq \Delta(z_{k+1}, w_{k+1}) - \frac{1}{\eta}(3D + 2\eta \max \|F_\infty\|) \max \|G_k\|$ if $D = \max\{\|w_{k+1} - z_{k+1}\|, \|z_{k+1} - z_k\|\}$ and $\mathcal{Z} = \mathbb{R}^n$.*

*Proof.* In this proof, "LHS" stands for "left-hand side". Due to Lemma 8, there exists

$$
\eta^2 \Delta(z_k, w_k) - \eta^2 \Delta(z_{k+1}, w_{k+1}) = \left( \eta^2 r^{tan}(z_k)^2 + \eta^2 \|F_\infty(z_k) - F_\infty(w_k)\|^2 \right)
$$

$$
- \left( \eta^2 r^{tan}(z_{k+1})^2 + \eta^2 \|F_\infty(z_{k+1}) - F_\infty(w_{k+1})\|^2 \right)
$$

$$
= \left( \|\eta F_\infty(z_k) + \eta c_k\|^2 + \eta^2 \|F_\infty(z_k) - F_\infty(w_k)\|^2 \right)
$$

$$
- \left( \|\eta F_\infty(z_{k+1}) + \eta c_{k+1}\|^2 + \eta^2 \|F_\infty(z_{k+1}) - F_\infty(w_{k+1})\|^2 \right)
\tag{16}
$$

Considering that $F_\infty$ is monotone and $L$-Lipschitz, and $\eta \in \left(0, \frac{1}{2L}\right)$, there exists

$$
-2 \langle \eta F_\infty(z_{k+1}) - \eta F_\infty(z_k), z_{k+1} - z_k \rangle \leq 0
\tag{17}
$$

$$
-2 \left( \frac{1}{4} \|z_{k+1} - w_{k+1}\|^2 - \|\eta F_\infty(z_{k+1}) - \eta F_\infty(w_{k+1})\|^2 \right) \leq 0
\tag{18}
$$

According to the OG algorithm, there exists $z_k - \eta F_\infty(w_k) - \eta G_k(w_k) - w_{k+1} \in N(w_{k+1})$ and $z_k - \eta F_\infty(w_{k+1}) - \eta G_k(w_{k+1}) - z_{k+1} \in N(z_{k+1})$. Hence,

$$
- \langle z_k - \eta F_\infty(w_k) - \eta G_k(w_k) - w_{k+1}, w_{k+1} - z_{k+1} \rangle \leq 0
\tag{19}
$$

$$
-2 \langle z_k - \eta F_\infty(w_{k+1}) - \eta G_k(w_{k+1}) - z_{k+1}, z_{k+1} - z_k \rangle \leq 0
\tag{20}
$$

Since $c(z_k) \in N(z_k)$, there exists

$$
- \langle \eta c(z_k), z_k - w_{k+1} \rangle \leq 0
\tag{21}
$$

$$
- \langle \eta c(z_k), z_k - z_{k+1} \rangle \leq 0
\tag{22}
$$

According to Lemma 8 and $z_k - \eta F_k(w_{k+1}) - z_{k+1} \in N_{\mathcal{Z}}(z_{k+1})$, $c_{k+1} \in \Pi_{N_{\mathcal{Z}}(z_{k+1})}(-F_\infty(z_{k+1}))$,

$$
-2 \langle \eta c(z_{k+1}) + \eta F_\infty(z_{k+1}), z_k - \eta F_\infty(w_{k+1}) - \eta G_k(w_{k+1}) - z_{k+1} \rangle \leq 0
\tag{23}
$$

$$
-2 \langle \eta c(z_{k+1}) + \eta F_\infty(z_{k+1}), -\eta c(z_{k+1}) \rangle \leq 0
\tag{24}
$$

there exists the following inequality:

Expression (16) + LHS of Inequality (17) + LHS of Inequality (18) + LHS of Inequality (19) +LHS of Inequality (20) + LHS of Inequality (21) + LHS of Inequality (22) + LHS of Inequality (23) +LHS of Inequality (24)

$$
\begin{aligned}
= & \left\| \frac{w_{k+1} - z_{k+1}}{2} + \eta F_\infty(w_k) - \eta F_\infty(z_k) \right\|^2 + \left\| \eta F_\infty(z_k) + \eta c(z_k) - z_k + \frac{w_{k+1} + z_{k+1}}{2} \right\|^2 \\
& + \| z_k - \eta F_\infty(w_{k+1}) - z_{k+1} - \eta c(z_{k+1}) \|^2 + \langle \eta G_k(w_k), w_{k+1} - z_{k+1} \rangle \\
& + 2 \langle \eta G_k(w_{k+1}), z_{k+1} - z_k \rangle + 2 \langle \eta G_k(w_{k+1}), \eta c(z_{k+1}) + \eta F_\infty(z_{k+1}) \rangle \\
\geq & - \eta \| w_{k+1} - z_{k+1} \| \max \|G_k\| - 2\eta \|z_{k+1} - z_k\| \max \|G_k\| - 4\eta^2 \max \|F_\infty\| \max \|G_k\| \\
\geq & - \eta \left( 3D_{\mathcal{Z}} + 4\eta \max \|F_\infty\| \right) \max \|G_k\|
\end{aligned}
$$

(25)

and there exists the following inequality if the game is unconstrained:

Expression (16) + LHS of Inequality (17) + LHS of Inequality (18) + LHS of Inequality (19) +LHS of Inequality (20) + LHS of Inequality (21) + LHS of Inequality (22) + LHS of Inequality (23) +LHS of Inequality (24)

$$
\begin{aligned}
\geq & \langle \eta G_k(w_k), w_{k+1} - z_{k+1} \rangle + 2 \langle \eta G_k(w_{k+1}), z_{k+1} - z_k \rangle + 2 \langle \eta G_k(w_{k+1}), \eta F_\infty(z_{k+1}) \rangle \\
\geq & - \eta D \max \|G_k\| - 2\eta D \max \|G_k\| - 2\eta^2 \max \|F_\infty\| \max \|G_k\| \\
\geq & - \eta \left( 3D + 2\eta \max \|F_\infty\| \right) \max \|G_k\|
\end{aligned}
$$

□

## C.4 Convergence rate of $\Delta(z_k, w_k)$

In the following lemma, we demonstrate the relationship between $r^{tan}_{(F,\mathcal{Z})}(w_{k+1})$ and $\Delta(z_k, w_k)$.

**Lemma 21** *For a game $\mathcal{G}$ converging to a monotone game with any closed convex set $\mathcal{Z} \subseteq \mathbb{R}^n$ and monotone and $L$-Lipschitz operator $F_\infty : \mathcal{Z} \to \mathbb{R}^n$ and $z_k, w_k, w_{k+1} \in \mathcal{Z}$, there exists*

$$
r^{tan}(w_{k+1}) \leq \sqrt{2} \left(2 + \eta L\right) \sqrt{\Delta(z_k, w_k)} + (2 + \eta L) \max \|G_k\|
$$

*If $G_k$ is $L_{G_k}$-Lipschitz and $G_t(z^*) = 0$, there exists*

$$
r^{tan}(w_{k+1}) \leq \sqrt{2} \left(2 + \eta L\right) \sqrt{\Delta(z_k, w_k)} + (1 + \eta L) \max \|G_k\| + L_{G_k} D
$$

*where $D = \min\{\max \|z_k - w_k\|, D_{\mathcal{Z}}\}$ under Assumption 4, and $D = D_{\mathcal{Z}}$ under other circumstances.*

*Proof.* Due to the OG algorithm and the non-expansiveness of the projection operator, there exists

$$
\begin{aligned}
\|z_k - w_{k+1}\| & \leq \|z_k - \Pi_{\mathcal{Z}} [z_k - \eta F_\infty(z_k)]\| + \|w_{k+1} - \Pi_{\mathcal{Z}} [z_k - \eta F_\infty(z_k)]\| \\
& \leq r^{tan}_{(\eta F, \mathcal{Z})}(z_k) + \|\Pi_{\mathcal{Z}} [z_k - \eta F_k(w_k)] - \Pi_{\mathcal{Z}} [z_k - \eta F_\infty(z_k)]\| \\
& \leq \eta r^{tan}_{(F,\mathcal{Z})}(z_k) + \|\eta F_\infty(w_k) - \eta F_\infty(z_k)\| + \|\eta G_k(z_k)\|
\end{aligned}
$$

According to Lemma 10, we have

$$
r^{tan}(z_k) \leq \frac{1 + \eta L}{\eta} \|z_k - w_{k+1}\| + \|F_\infty(z_k) - F_\infty(w_k)\| + \max\{\|G_k\|\}
$$

Hence, there exists

$$
\begin{aligned}
r^{tan}(w_{k+1}) & \leq (1 + \eta L) r^{tan}_{(F,\mathcal{Z})}(z_k) + (2 + \eta L) \|F_\infty(w_k) - F_\infty(z_k)\| + (2 + \eta L) \max \|G_k\| \\
& \leq \sqrt{2} \left(2 + \eta L\right) \sqrt{r^{tan}_{(F,\mathcal{Z})}(z_k)^2 + \|F_\infty(w_k) - F_\infty(z_k)\|^2} + (2 + \eta L) \max \|G_k\| \\
& = \sqrt{2} \left(2 + \eta L\right) \sqrt{\Delta(z_k, w_k)} + (2 + \eta L) \max \|G_k\|
\end{aligned}
$$

If $G_k$ is $L_{G_k}$-Lipschitz, there exists

$$r^{tan}(w_{k+1}) \leq \left\| \frac{z_k - w_{k+1}}{\eta} + F_\infty(w_{k+1}) - F_\infty(w_k) \right\| + L_{G_k} D$$

$$\leq \frac{1}{\eta} \|z_k - w_{k+1}\| + \|F_\infty(w_{k+1}) - F_\infty(z_k)\| + \|F_\infty(z_k) - F_\infty(w_k)\| + L_{G_k} D$$

$$\leq \frac{1 + \eta L}{\eta} \|z_k - w_{k+1}\| + \|F_\infty(z_k) - F_\infty(w_k)\| + L_{G_k} D$$

Hence, there exists

$$r^{tan}(w_{k+1}) \leq (1 + \eta L) \, r^{tan}_{(F,\mathcal{Z})}(z_k) + (2 + \eta L) \|F_\infty(w_k) - F_\infty(z_k)\| + (1 + \eta L) \|G_k(z_k)\|$$

$$+ L_{G_k} D$$

$$\leq \sqrt{2} \, (2 + \eta L) \sqrt{r^{tan}_{(F,\mathcal{Z})}(z_k)^2 + \|F_\infty(w_k) - F_\infty(z_k)\|^2}$$

$$+ (1 + \eta L) \|G_k(z_k)\| + L_{G_k} D$$

$$\leq \sqrt{2} \, (2 + \eta L) \sqrt{\Delta(z_k, w_k)} + (1 + \eta L) \max \|G_k\| + L_{G_k} D$$

$\square$

Now we combine all the results above in this section in Theorem 6 for the proof of Theorem 4.

**Theorem 6** *For a game $\mathcal{G}$ converging to a monotone game, under Assumption 2 if $\eta \in \left(0, \frac{1}{\sqrt{6}L}\right)$, or under Assumption 3 if $\eta \in \left(0, \frac{1}{\sqrt{6L^2 + 4LL_G + 2L_G^2}}\right)$, or under Assumption 4 if $\eta \in \left(0, \min\left\{\frac{1}{2(L+L_G)}, \frac{1}{4L_G}\right\}\right)$, $\sqrt{\Delta(z_{T_1+T_2}, w_{T_1+T_2})}, r^{tan}(z_{T_1+T_2}) \leq \frac{C_1}{\sqrt{T_2}} + \sqrt{C_2 \sum_{k=T_1}^\infty \max \|G_k\|}$. Under Assumption 2, if $\eta \in \left(0, \frac{1}{\sqrt{6}L}\right)$,*

$$r^{tan}(w_{T_1+T_2+1}) \leq \frac{\sqrt{2} \, (2 + \eta L) C_1}{\sqrt{T_2}} + \sqrt{2} \, (2 + \eta L) \sqrt{C_2 \sum_{k=T_1}^\infty \max \|G_k\|}$$

$$+ (2 + \eta L) \max \|G_{T_1+T_2}\|$$

$$C_1 = \frac{1}{\eta} \left( \frac{4 + 20\eta^4 L^4}{1 - 6\eta^2 L^2} \max_{t \in \mathbb{N}} \|z_t - z^*\|^2 + \frac{4\eta^2 L^2 \left(5\eta^2 L^2 + 6\right)}{1 - 6\eta^2 L^2} \max_{t \in \mathbb{N}} \|w_t - z_t\|^2 \right.$$

$$\left. + \frac{4\eta^2 \left(9 + 4\eta^2 L^2 - 20\eta^4 L^4\right)}{1 - 6\eta^2 L^2} E_{mk^2} + \frac{4\eta D_\mathcal{Z} \left(4 + 20\eta^4 L^4\right)}{1 - 6\eta^2 L^2} E_{mk} \right)^{\frac{1}{2}}$$

$$C_2 = \frac{3 D_\mathcal{Z} + 4\eta \max \|F_\infty\|}{\eta}$$

*Under Assumption 3,*

$$r^{tan}(w_{T_1+T_2+1}) \leq \frac{\sqrt{2} \, (2 + \eta L) C_1}{\sqrt{T_2}} + \sqrt{2} \, (2 + \eta L) \sqrt{C_2 \sum_{k=T_1}^\infty \max \|G_k\|}$$

$$+ (2 + \eta L) \max \|G_T\| + L_{G_T} D_\mathcal{Z}$$

$$C_1 = \frac{1}{\eta} \left( \frac{4(1 - 4\eta^2 L^2) + 2\eta^2 L^2 \left(\eta^2 (L + L_G)^2 + 8 + 8\eta^2 L^2\right)}{1 - 6\eta^2 L^2 - 4\eta^2 L L_G - 2\eta^2 L_G^2} \max_{t \in \mathbb{N}} \|z_t - z^*\|^2 + \max_{t \in \mathbb{N}} \|w_t - z_t\|^2 \right.$$

$$\cdot \left( \frac{2\eta^2 (L + L_G)^2 \left(4 + 2\eta^2 L^2 \left(8\eta^2 L^2 + \eta^2 (L + L_G)^2\right)\right)}{(1 - 6\eta^2 L^2 - 4\eta^2 L L_G - 2\eta^2 L_G^2)(1 - 4\eta^2 L^2)} + \frac{2\eta^2 L^2 \left(8 + 8\eta^2 L^2 + \eta^2 (L + L_G)^2\right)}{1 - 4\eta^2 L^2} \right)$$

$$+ \frac{8\eta D_\mathcal{Z}(1 - 4\eta^2 L^2) + 4\eta^3 D_\mathcal{Z} L^2 \left(\eta^2 (L + L_G)^2 + 8 + 8\eta^2 L^2\right)}{1 - 6\eta^2 L^2 - 4\eta^2 L L_G - 2\eta^2 L_G^2} E_{mk} + 4\eta^2 E_{mk^2} \left(9 + 8\eta^2 L^2\right)$$

$$+ \frac{2\eta^2 L^2 \left(8 + 8\eta^2 L^2 + \eta^2 \left(L + L_G\right)^2\right)}{1 - 4\eta^2 L^2} + \frac{2\eta^2 \left(L + L_G\right)^2 \left(4 + 2\eta^2 L^2 \left(8\eta^2 L^2 + \eta^2 \left(L + L_G\right)^2\right)\right)}{\left(1 - 6\eta^2 L^2 - 4\eta^2 L L_G - 2\eta^2 L_G^2\right)\left(1 - 4\eta^2 L^2\right)}\right)\right)^{\frac{1}{2}}$$

$$C_2 = \frac{3D_{\mathcal{Z}} + 4\eta \max \|F_\infty\|}{\eta}$$

*Under Assumption 4,*

$$r^{tan}(w_{T_1+T_2+1}) \le \frac{\sqrt{2}\left(2 + \eta L\right) C_1}{\sqrt{T_2}} + \sqrt{2}\left(2 + \eta L\right)\sqrt{C_2 \sum_{k=T_1}^{\infty} \max \|G_k\|}$$
$$+ \left(1 + \eta L\right) \max \|G_T\| + L_{G_T} D$$

$$C_1 = \frac{1}{\eta}\left(\frac{4 + E_G \eta^2 \left(L + L_G\right)^2 \left(4 + 3\eta^2 L^2\right)}{1 - 4\eta L_G - E_G E_{G2} \eta^2 \left(L + L_G\right)^2} E_{G2} \max_{t \in \mathbb{N}} \|z_t - z^*\|^2\right.$$
$$+ \left(\frac{4 E_{G2} + E_G E_{G2} \eta^2 \left(L + L_G\right)^2 \left(4 + 3\eta^2 L^2\right)}{1 - 4\eta L_G - E_G E_{G2} \eta^2 \left(L + L_G\right)^2} + 4 + 3\eta^2 L^2\right)\left(\eta^2 \left(L + L_G\right)^2 E_G\right) \max_{t \in \mathbb{N}} \|w_t - z_t\|^2$$
$$\left. + 2\eta^2 D_0^2 \sum_{k=1}^{T} L_{G_{k-1}}\left(2L + L_{G_{k-1}}\right)\right)^{\frac{1}{2}}$$

$$C_2 = \frac{3D + a\eta \max \|F_\infty\|}{\eta}$$

*where $D_0 = \max_{t \in \mathbb{N}}\{\|z_t - z^*\|, \|z_t - w_t\|\}$, $D = \max\{\|w_{k+1} - z_{k+1}\|, \|z_{k+1} - z_k\|\}$, $k \in \mathbb{N}$, and for unconstrained cases $a = 2$ and for constrained cases $a = 4$.*

*Proof.* According to Lemma 5, there exists $t^* \in (T_1, T_1 + T_2] \cap \mathbb{N}^*$ so that $\Delta(z_{t^*}, w_{t^*})^2 = O(1/T_2)$. Since $T_1 < t^*$, according to Theorem 3, there exists

$$\Delta(z_{T_1+T_2}, w_{T_1+T_2}) \le \Delta(z_{t^*}, w_{t^*}) + \sum_{k=T_1}^{\infty} \frac{3D_{\mathcal{Z}} + 4\eta \max \|F_\infty\|}{\eta} \max \|G_k\|$$

under any one of Assumptions 2 and 3, and there exists

$$\Delta(z_{T_1+T_2}, w_{T_1+T_2}) \le \Delta(z_{t^*}, w_{t^*}) + \sum_{k=T_1}^{\infty} \frac{3D + a\eta \max \|F_\infty\|}{\eta} \max \|G_k\|$$

under Assumption 4 with unconstrained games and $a = 2$. Considering that all $z_i, w_i, i \in \mathbb{N}$ are in a bounded convex set due to Lemma 19, we have $a = 4$ under Assumption 4 in a constrained game. The theorem is obtained by directly combining the two equations above with Lemma 5 and Lemma 19. $\qquad\square$

## C.5 PROOF OF THEOREM 4

Cases where $T < 3$ are trivial with Big $O$ notation since $a(x) = O(b(x))$ is true if $x$ belongs to a limited set, $\forall a(x) \ge 0$ and $b(x) > 0$. Suppose $T = T_1 + T_2 + 1, T_1, T_2 \in \mathbb{N}^*$ for $T \ge 3$ and $T_1 = T_2 + 2$ or $T_2 + 1$. According to Theorem 6, we have

$$r^{tan}(w_T) = r^{tan}(w_{T_1+T_2+1}) \le \frac{\sqrt{2}\left(2 + \eta L\right) C_1}{\sqrt{T_2}} + \sqrt{2}\left(2 + \eta L\right)\sqrt{C_2 \sum_{k=T_1}^{\infty} \max \|G_k\|}$$
$$+ \left(2 + \eta L\right) \max \|G_{T_1+T_2}\|$$

where $C_1, C_2 > 0$ under Assumptions 2 and 3. Hence,

$$r^{tan}(w_T) = O\left(\frac{1}{\sqrt{T_2}}\right) + O\left(\sqrt{\sum_{k=T_1}^{\infty} \max \|G_k\|}\right) + O\left(\max \|G_{T-1}\|\right)$$
$$= O\left(\max\left\{\frac{1}{\sqrt{T}}, \sqrt{\sum_{k=T/2}^{\infty} \max \|G_k\|}, \max \|G_T\|\right\}\right)$$

Also, we have

$$
\begin{aligned}
r^{tan}(w_T) = r^{tan}(w_{T_1+T_2+1}) \leq & \frac{\sqrt{2}\,(2+\eta L)\,C_1}{\sqrt{T_2}} + \sqrt{2}\,(2+\eta L)\,\sqrt{C_2 \sum_{k=T_1}^{\infty} \max \|G_k\|} \\
& + (1+\eta L)\max\|G_{T-1}\| + L_{G_{T-1}} D_0
\end{aligned}
$$

where $C_1, C_2 > 0$ under Assumption 4 and $D_0$ is $D$ in Theorem 6. $\forall \epsilon > 0$, $\exists N_0$, $\forall N > N_0$, $\left| \prod_{k=N}^{\infty} (1 + 4\eta L_{G_k}) - 1 \right| < \epsilon$ and $L_{G_N} < \epsilon$ so that $1 - 4\eta L_G - E_G E_{G2}\eta^2 (L + L_G)^2 > 0$ if $N_0$ were the initial time 0. Since $\forall N_0 > 0$, $T = O(T - N_0)$ for $T > N_0$, $(T + N_0)/2 > T/2$ and $E_G < 4$, we have

$$
\begin{aligned}
r^{tan}(w_T) = & O\left(\frac{1}{\sqrt{T_2}}\right) + O\left(\sqrt{\sum_{k=T_1}^{\infty} \max\|G_k\|}\right) + O\left(\max\|G_T\|\right) + O\left(L_{G_{T-1}} D_0\right) \\
= & O\left(\max\left\{\frac{1}{\sqrt{T}}, \sqrt{\sum_{k=T/2}^{\infty} \max\|G_k\|}, \max\|G_T\|, L_{G_{T-1}}\right\}\right)
\end{aligned}
$$

Then, according to Lemma 3, $\frac{P_{\mathcal{G},D}(z)}{D} \leq r_{\mathcal{G}}^{tan}(z)$ and $\frac{T_{\mathcal{G},D}(z)}{\sqrt{N}D} \leq r_{\mathcal{G}}^{tan}(z)$. Hence, we have

$$
\max\left\{ r^{tan}(w_T), \frac{T_{\mathcal{G},D}(w_T)}{\sqrt{N}D}, \frac{P_{\mathcal{G},D}(w_T)}{D} \right\} = r^{tan}(w_T)
$$

Therefore,

$$
\begin{aligned}
& \max\left\{ r^{tan}(w_T), \frac{T_{\mathcal{G},D}(w_T)}{\sqrt{N}D}, \frac{P_{\mathcal{G},D}(w_T)}{D} \right\} \\
= & O\left(\max\left\{\frac{1}{\sqrt{T}}, \sqrt{\sum_{k=T/2}^{\infty} \max\|G_k\|}, \max\|G_T\|\right\}\right)
\end{aligned}
$$

under Assumptions 2 and 3, including $D = D_{\mathcal{Z}} > 0$, while

$$
\begin{aligned}
& \max\left\{ r^{tan}(w_T), \frac{T_{\mathcal{G},D}(w_T)}{\sqrt{N}D}, \frac{P_{\mathcal{G},D}(w_T)}{D} \right\} \\
= & O\left(\max\left\{\frac{1}{\sqrt{T}}, \sqrt{\sum_{k=T/2}^{\infty} \max\|G_k\|}, \max\|G_T\|, L_{G_{T-1}}\right\}\right)
\end{aligned}
$$

under Assumption 4. □

