# OpenReview forum: "Classic but Everlasting: Traditional Gradient-Based Algorithms Converge Fast Even in Time-Varying Multi-Player Games"
_ICLR.cc/2025/Conference — ICLR 2025 Oral_

### Official Review · Reviewer_Gqyb · 2024-10-27

**Soundness:** 3
**Presentation:** 3
**Contribution:** 3
**Rating:** 8
**Confidence:** 3

**Summary:**

This paper proves last-iterate convergence rates of extra-gradient (EG) and optimal-gradient (OG) algorithms for multi-player smooth games with bounded actions and with time-varying cost functions. The time-varying game is assumed to converge to a smooth monotone game with time, i.e. the perturbations decay to zero eventually.

**Strengths:**

While previous convergence results for EG and OG are mainly provided for time-invariant games and time-varying games that are restricted to be two-player, bilinear, zero-sum, this paper pushes the boundary and shows fast convergence results of EG and OG algorithms for multi-player, time-varying games that eventually converge to smooth monotone games and with convex constraint set for actions. In general, I think it is a strong submission and the results are quite interesting, espcially that the author(s) use tangent residual to measure proximity to Nash to overcome challenges could appear when using gap functions.

**Weaknesses:**

1. (Starting line 432) The numerical experiments section
- The author(s) claim that they provide results for general multi-player games beyond two-player zero-sum games. I assume by this claim they mean multi-player general-sum games. However, the experiments only show results for zero-sum cost functions. I think the author(s) should also show results for general non-zero-sum games involving multiple players in order to make their point. Otherwise, I would like to hear more elaboration on that end.
- The perturbation added to the two players seems to be identical; is that something required for the results to hold? If not, I would encourage the author(s) to remove the restriction in their experimental results.
- The initial z0 picked in the experiments seems to be very non-arbitrary---both players start from exactly the same point. Could the author(s) elaborate on this? Ideally, the empirical results can be much strengthened by a Monte-Carlo study with many different initial conditions and report of statistical results.

2. Other minor aspects: on a high level, quite some notations and concepts are used in the paper without giving definitions. I would appreciate a clearer presentation.
- Line 131: variables in the game tuple undefined; game itself undefined; constrained set Z undefined
- Line 141: variational equality → variational inequality
- Line 171: existence of at least one Nash equilibrium better be stated explicitly as an  assumption
- Eq (1):  Euclidean projection operator undefined
- Line 226: Dr(z) undefined

**Questions:**

1. Even two-player zero-sum bilinear games are not strictly monotone; how restrictive it is to assume that the game converges to a smooth monotone game eventually?
2. If Nash equilibria exist, can these two learning algorithms guarantee to only converge to Nash equilibria? If not, what else assumptions and/or treatment are needed?

---

### Official Review · Reviewer_KZkR · 2024-11-04

**Soundness:** 4
**Presentation:** 3
**Contribution:** 3
**Rating:** 8
**Confidence:** 3

**Summary:**

In this paper, the authors present the last-iterate convergence of EG and OG algorithms in bounded and constrained time-varying, multiplayer games that converges to a monotone game. The convergence results derived for the time-varying games are similar to known tight results on corresponding time-invariant games.

In specific, the author proved that EG and OG with a constant learning rate both converge to a Nash equilibrium at the rate of $\mathcal{O}(1/\sqrt{T})$ under three conditions (Assumption 1 (general convex bounded), 2 ($L_G$-smooth perturbing), 3 ($L_G$-smooth perturbing+$G_t(z^*)=0$)) respectively.

**Strengths:**

- The paper is well-motivated, with the introduction and related work sections effectively contextualizing recent advances in time-varying games. The authors clearly articulate how this work improves upon existing results in the field.
- The paper is well-organized: it begins with foundational concepts, including definitions of the game, learning algorithms, the measure of proximity to Nash Equilibrium (tangent residual) and its properties, as well as assumptions about the games. The authors then proceed with separate analyses of the convergence of EG and OG methods, systematically outlining the proof techniques and presenting each step in detail.
- While I did not examine every detail of the proofs, the mathematical reasoning appears clear and correct.

**Weaknesses:**

- My primary concern is that the contribution of this paper may be perceived as an incremental extension of the method from [1] to time-varying games.
In specific, for proofs of last-iterate convergence of both EG and OG, the author seems to follow the same proof procedure as [1] does but adding additional analysis for the time-varying $G_t$'s which results in additional terms in the bounds.
For example, this paper's Lemma 4 argues best-iterate convergence with rate $\mathcal{O}(1/ \sqrt{T})$ of tangent residual, and this corresponds exactly to Lemma 5 of [1], while Lemma 13 and 15 corresponds to Lemma 3 and 4 of [1] respectively. Moreover, Theorem 1 which establishes the monotonicity of the tangent residual, corresponds to Theorem 2 of [1].

I acknowledge the authors' careful analysis of the perturbation terms. However, could you highlight and summarize the main proof techniques that are novel and distinct from those used in [1]? This additional clarification would be greatly appreciated, and discovering additional novel technical contributions would very likely lead me to improve my evaluation.




[1] Yang Cai, Argyris Oikonomou, and Weiqiang Zheng. Finite-time last-iterate convergence for learning in multi-player games. In S. Koyejo, S. Mohamed, A. Agarwal, D. Belgrave, K. Cho, and A. Oh (eds.), Advances in Neural Information Processing Systems, volume 35, pp. 33904–33919. Curran Associates, Inc., 2022. URL https://proceedings.neurips.cc/paper_files/paper/2022/file/
db2d2001f63e83214b08948b459f69f0-Paper-Conference.pdf.

**Questions:**

- For assumption 3, $z^*$ is the NE right? Then would $G_t(z^*)=0$ be a very strong assumption? It makes senses to me that $G_{\infty}(z^*)=0$, but what's the natural justification of assuming that  $G_t(z^*)=0$ for all $t$?
- Also, for assumption 2 and 3, do you assume that $G_t$ is $L(G_t)$-Lipschitz for all $t$?
- I believe it's a typo, but in the first line of the conclusion, you want to say "bounded and constrained cases'' instead of "bounded and unconstrained cases" right?
- For lots of your theorem statements, you include terms like $\max ||G_{t^*}||$, $\max ||G_k||$, $\max ||F_\infty||$, can you specify what $\max$ you are taking respect to? Like $\max_{z\in \mathcal{Z}} ||G_{t^*}(z)||$? Since it's quite confusing right now.

---

### Official Review · Reviewer_BAfq · 2024-11-07

**Soundness:** 4
**Presentation:** 2
**Contribution:** 3
**Rating:** 8
**Confidence:** 3

**Summary:**

The authors study time-varying games in, both, bounded and unconstrained strategy spaces. In particular, they are interested in the performance of Extra Gradient and Optimistic Gradient algorithms in sequences of games that converge to some smooth monotone (also known as diagonally concave) game, i.e., the pseudogradient of the player's payoff functions is monotone and L-Lipschitz. They prove that, under bounded accumulated perturbations assumptions, both algorithms converge to the set of Nash equilibria of the monotone game at a rate approximately O(sqrt(T)^{-1}) with constant learning rates.

**Strengths:**

Although I am not familiar with the majority of the literature, I believe the class of time-varying games studied in this work is rich and interesting. Therefore, the novelty of this work should be appreciated.

**Weaknesses:**

1. Could the authors improve the theorem statements? Although I appreciate the accuracy of the mathematical formalism, the text has not been refined adequately to improve the clarity of the statements. I suggest either expressing some of the quantities involved in Theorems 1 through 4 in English or/and predefining some of the maximization functions before the actual statement.
2. There exist multiple typos in the manuscript. A careful read is advised. I mention here the ones I find the most hurtful for clarity:
* In lines 85 and 88 z^{*} is used before it is defined.
* In line 163 \lim_{t \to \infty} G is not defined in Definition 2.

**Questions:**

Kindly refer to the weaknesses.

---

### Meta-Review · Area_Chair_Ttu8 · 2024-12-21

**Metareview:**

The paper focuses on time-varying games settings. Specifically, the authors are interested in last iterate convergence guarantees of Extra Gradient and Optimistic Gradient algorithms for time varying games that converge  to some smooth monotone (also known as diagonally concave) game. It is proved that, under bounded accumulated perturbations assumptions, both algorithms converge to the set of Nash equilibria of the monotone game at a rate approximately O(sqrt(T)^{-1}), using constant learning rates. All the reviewers were positive about this work and so is the AC. We recommend acceptance.

**Additional Comments On Reviewer Discussion:**

The reviewers believe the paper is worth acceptance and have high scores.

---

### Decision · Program_Chairs · 2025-01-22

Accept (Oral)